# The challenge of population aging for mitigating deaths from PM$_{2.5}$ air pollution in China

Fangjin Xu[1,2,3], Qingxu Huang [1,2,4] ✉, Huanbi Yue[5], Xingyun Feng[1,2], Haoran Xu [1,4], Chunyang He [1,6,7,8], Peng Yin[9] & Brett A. Bryan [10]

Estimating the health burden of air pollution against the background of population aging is of great significance for achieving the Sustainable Development Goal 3.9 which aims to substantially reduce the deaths and illnesses from air pollution. Here, we estimated spatiotemporal changes in deaths attributable to PM$_{2.5}$ air pollution in China from 2000 to 2035 and examined the drivers. The results show that from 2019 to 2035, deaths were projected to decease 15.4% (6.6%–20.7%, 95% CI) and 8.4% (0.6%–13.5%) under the SSP1-2.6 and SSP5-8.5 scenario, respectively, but increase 10.4% (5.1%–20.5%) and 18.1% (13.0%–28.3%) under SSP2-4.5 and SSP3-7.0 scenarios. Population aging will be the leading contributor to increased deaths attributable to PM$_{2.5}$ air pollution, which will counter the positive gains achieved by improvements in air pollution and healthcare. Region-specific measures are required to mitigate the health burden of air pollution and this requires long-term efforts and mutual cooperation among regions in China.

Fine particulate matter (PM$_{2.5}$) has become one of the most ubiquitous air pollutants in China[1–3]. Exposure to PM$_{2.5}$ air pollution has adverse effects on human health, causing premature death via diseases such as pulmonary and cardiovascular disease[4,5]. Despite implementing a series of strict policies to control PM$_{2.5}$ air pollution[6], the number of deaths attributable to PM$_{2.5}$ air pollution (DAPP) in China has increased 36%, from 0.7 million to nearly 1 million, between 2000 and 2017[7]. In parallel, China's population is aging which is further exacerbating the negative health effects of air pollution[8–10]. China is experiencing particularly acute population aging, with the proportion of older adults growing from 7.0% to 12.6% from 2000-2019, rising to 17.1% and 26.3% by 2030 and 2050, respectively[11], and peaking at 437 million people by 2051[12]. The United Nations

SDG3.9 states that "people of all ages need to live healthy lives and that a substantial reduction in air pollution-related deaths needs to be achieved by 2030"[13]. Understanding the health effects of air pollution in the context of population aging in China is of great significance for formulating policies to promote the well-being of older adults, e.g., establishing a pre-warning system for reducing exposure to heavily polluted air, increasing financial investment to alleviate their economic burden for treating related diseases. The insights gained can help other countries deal with the health burden of air pollution in the context of population aging and promote progress towards SDG 3.9.1 (reducing mortality rate attributed to household and ambient air pollution) and SDG 11.6.2 (reducing annual mean levels of fine particulate matter).

[1]State Key Laboratory of Earth Surface Processes and Resource Ecology, Beijing Normal University, Beijing 100875, China. [2]School of Natural Resources, Faculty of Geographical Science, Beijing Normal University, Beijing 100875, China. [3]College of Urban and Environmental Sciences, Peking University, Beijing 100871, China. [4]Faculty of Geographical Science, Beijing Normal University, Beijing 100875, China. [5]School of International Affairs and Public Administration, Ocean University of China, Qingdao 266100, China. [6]Key Laboratory of Environmental Change and Natural Disasters, Ministry of Education, Beijing Normal University, Beijing 100875, China. [7]Academy of Disaster Reduction and Emergency Management, Ministry of Emergency Management and Ministry of Education, Beijing 100875, China. [8]Academy of Plateau Science and Sustainability, People's Government of Qinghai Province and Beijing Normal University, Xining, China. [9]National Center for Chronic and Non-communicable Disease Control and Prevention, Chinese Center for Disease Control and Prevention, Beijing 100050, China. [10]School of Life and Environmental Sciences, Deakin University, Melbourne VIC3125, Australia. ✉e-mail: qxhuang@bnu.edu.cn

The number of DAPP is determined by four main factors: age structure, total population, $PM_{2.5}$ concentration, and disease mortality[14]. Changes in these four factors are influenced by both climate change and socioeconomic change. For example, the changes in climate conditions such as temperature, humidity, and air pressure can directly affect the secondary transformation and diffusion of $PM_{2.5}$[15,16], thereby changing the atmospheric concentration of $PM_{2.5}$. Socioeconomic development affects the emission of pollutants and $PM_{2.5}$ concentration, as well as the demographic characteristics and the level of available healthcare, which affect the proportion of older adults in the population and the mortality rate of diseases[17,18]. Several studies have investigated the relationship between $PM_{2.5}$ air pollution and the older adult population in China, mainly based on historical data to analyze the impact of population aging on a specific disease[19–21] (e.g., respiratory disease or heart disease). For example, Chi et al. analyzed the impact of air pollution on older patients with chronic obstructive pulmonary disease and found that increased $PM_{2.5}$ air pollution from heating during the winter months was associated with a reduction in forced expiratory volume[21]. Several studies have examined historical relationships between population aging and DAPP. For example, Liu et al. estimated the number of deaths caused by air pollution and concluded that population aging caused an additional 950,000 deaths from 2004 to 2017, delaying the reduction of the health burden of $PM_{2.5}$ air pollution[22]. A few studies have estimated future changes of DAPP, with estimates based upon certain climatic or socioeconomic assumptions. For example, Yue et al. estimated China's DAPP in 2030 under different $PM_{2.5}$ concentration scenarios, but ignored changes in demographics and disease mortality[7]. Wang et al. estimated the health burden of DAPP in 2020 and 2030 based on air quality scenarios and population trends under the Shared Socioeconomic Pathways (SSPs)[23]. Although the study estimated the impact of population aging and other factors on DAPP in China based on future socioeconomic changes, it did not consider the changes in $PM_{2.5}$ concentration in the context of climate change. Hence, there is an urgent need for a comprehensive and detailed quantification and projection of DAPP trends in China which ties these complex components together.

The sixth Coupled Model Intercomparison Project (CMIP6) considers socioeconomic development as well as climate change factors and can reflect the interaction between them, providing the possibility to effectively predict DAPP in the future. CMIP6 combines representative concentration pathways (RCPs) with shared socioeconomic pathways (SSPs). Specifically, RCPs are multi-model global scenarios of greenhouse gas concentration trajectories. They use the radiative forcing per unit area at the end of the century to represent the climate impact of future greenhouse gas concentrations, covering a variety of climate mitigation levels and providing the basis for the simulation of $PM_{2.5}$ in different climate change scenarios[16]. SSPs are socioeconomic development scenarios related to greenhouse gas emissions that take socioeconomic aspects such as economy, population, technology, lifestyle, policies, and institutions into account[24], making it possible to study the socioeconomic factors affecting DAPP. In addition, Yin et al.[25] estimated disease mortality associated with air pollution across provinces in China from 1990 to 2017, providing an important database for estimating differences in DAPP at the provincial scale. Decomposition analysis[7] can transform the nonlinear relationship between DAPP and its various drivers into a simple linear relationship, which takes into account the interaction between the various factors.

In this work, we estimate the temporal and spatial patterns of DAPP based on the CMIP6 dataset, provincial-level disease mortality data, and decomposition analysis, and investigate the impact of population aging in China under alternative future scenarios considering both socioeconomic development and climate change. We first estimated and verified the spatiotemporal pattern of DAPP in China from 2000 to 2019 based on the comparable risk assessment framework. Then, we estimated the changes in DAPP in China in different scenarios from 2020 to 2035 by combining the RCPs and SSPs. RCPs are multi-model global scenarios of greenhouse gases that provide the basis for future climate change scenarios. Based on the results, the influence of population aging as well as other driving factors (e.g., population growth, change in $PM_{2.5}$ concentrations and disease mortalities) were evaluated via decomposition analysis. The differences across regions are discussed and policy recommendations are made for improving public health. Here, we show that the trend in DAPP in China during 2019−2035 is projected to decease 15.4% (6.6%–20.7%, 95% CI) and 8.4% (0.6%–13.5%) under the SSP1-2.6 and SSP5-8.5 scenario, respectively, but increase 10.4% (5.1%–20.5%) and 18.1% (13.0%–28.3%) under SSP2-4.5 and SSP3-7.0 scenarios. Overall, population aging is the most important driver of DAPP growth from 2019 to 2035. In the *sustainability* (SSP1-2.6) and *fossil-fueled development* (SSP5-8.5) scenarios, population aging was the only factor resulting in increases in DAPP. In the future, the prevention and control of DAPP will require more effort from countries affected by air pollution to achieve the UN Sustainable Development Goals in the context of global aging. It is necessary to pay more attention to the health problems caused by population aging, increase healthcare investment for older adults, separate heavy industry from where people (especially older adults) live, and take more targeted measures to avoid the exposure of the older adults to air pollution. In addition, immediate action is needed to improve air quality, adhere to a sustainable development path, and avoid scenarios involving regional competition and the burning of fossil fuels.

## Results
### Trends in driving factors
From 2000 to 2019, China's population increased by 9.9%, yet the projected population (2019−2035) varied under different scenarios (Fig. 1). In the *middle of the road* scenario (SSP2-4.5) and the *regional rivalry* scenario (SSP3-7.0), the population grew from 2019, increasing by 3.0% and 5.7% by 2035, respectively[26]. In the *sustainability* (SSP1-2.6) and *fossil-fueled development* (SSP5-8.5) scenarios, the population remains steady. The percentage of older adults rose in all four scenarios, especially in SSP5-8.5, SSP1-2.6 and SSP2-4.5, where it was projected to increase by 93.4%, 92.4% and 92.4% from 2019−2035, respectively. In SSP3-7.0, the percentage of older adults increased by 79.3%.

The population-weighted $PM_{2.5}$ concentration in China increased by 43.8% from 2000 to 2019, while 2010 was a turning point, with an observed a decrease of $0.7\,\mathrm{ug\,m^{-3}}$ (1.6%) between 2010 and 2019 (Fig. 1). The estimated trends of the population-weighted $PM_{2.5}$ concentration revealed differences between the four scenarios (Fig. 1). Specifically, only SSP3-7.0 saw an increase, with the population-weighted $PM_{2.5}$ concentration reaching $43.3\,\mathrm{ug\,m^{-3}}$ by 2035. In the other three scenarios, the population-weighted $PM_{2.5}$ concentration was projected to decrease (2019−2035), with SSP1-2.6 experiencing the greatest decrease of 47.4%, followed by SSP2-4.5 (28.6%) and SSP5-8.5 (37.1%).

Most age-standardized death rates of diseases in different scenarios demonstrated a downward trend (Fig. 1). With the joint effects of socioeconomic development and improvement in health care, most death rates of diseases exhibited a distinct decrease, especially in the *sustainability* scenario (SSP1-2.6) and the *fossil-fueled development* scenario (SSP5-8.5). For instance, in SSP1-2.6, the age-standardized death rates of COPD, stroke and LRIs (2019−2035) decreased by 39.6%, 54.6% and 56.9%, respectively.

### DAPP trends in Chinaj
From 2019 to 2035, DAPP in China exhibited two contrasting trends for two stages in the four scenarios (Fig. 2a). From 2019−2025, annual DAPP under most scenarios was projected to decline, with decreases of 131.8 (86.6-140.8), 30.3 (15.7-40.8), and 118.6 (73.3-127.6) thousand

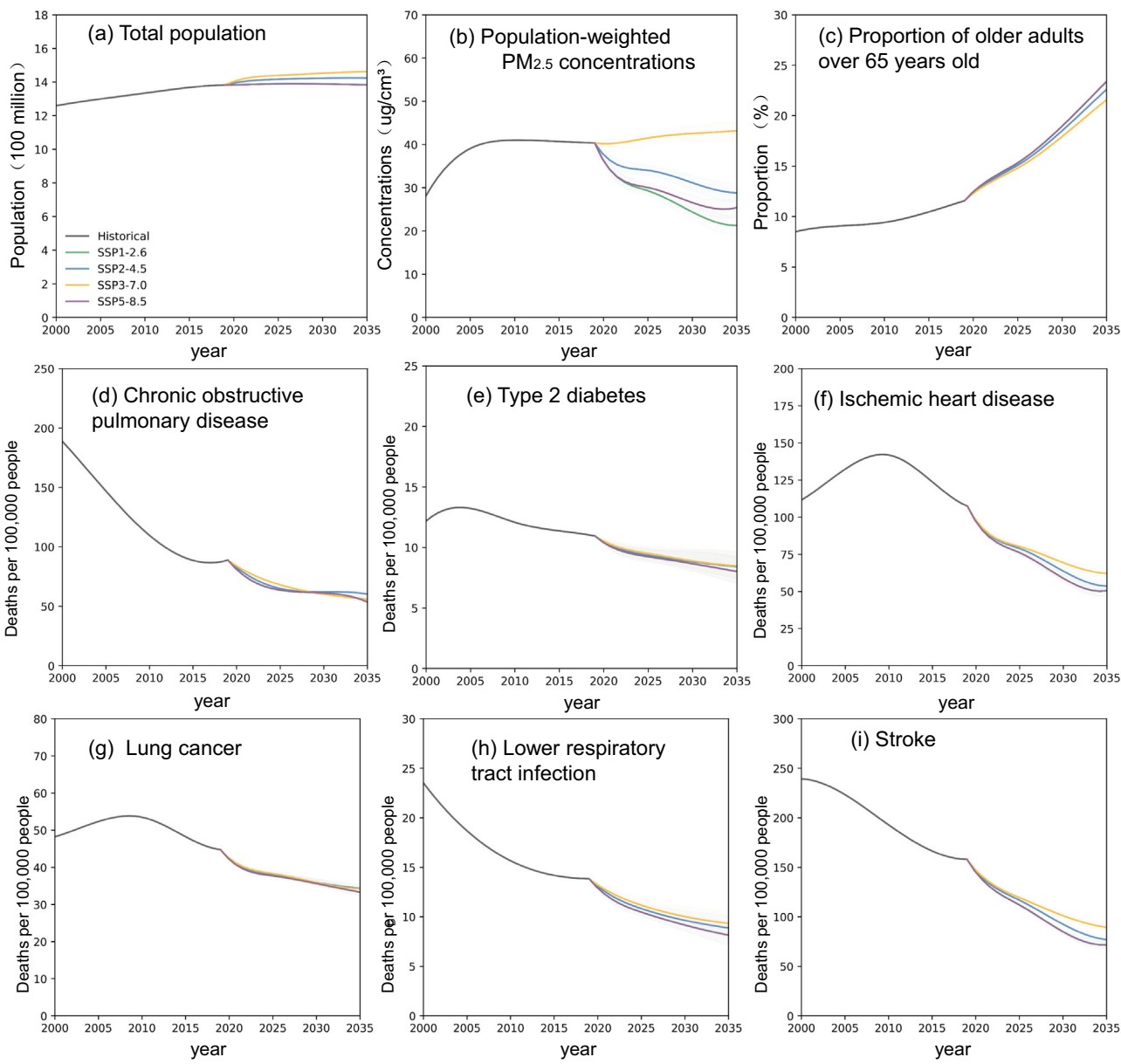

**Fig. 1 | Projected change in driving factors from 2000–2035. a–c** The trends of the total population, population-weighted $PM_{2.5}$ concentrations and the proportion of older adults (>65 years). **d–i** Age-standardized disease mortality. Note that the age structure of the sixth census in China was used as the standard age structure[65]. We used 5-year smoothed time intervals to visualize the results.

[parentheses contain the 95% confidence intervals] under SSP1-2.6, SSP2-4.5 and SSP5-8.5, respectively. Only in SSP3-7.0 scenarios, DAPP was projected to grow with increase of 9.7 (−0.1-34.0) thousand. Conversely, from 2025–2035, annual DAPP in China was projected to grow in most scenarios, with increases of 135.7 (56.4-209.7), 173.9 (95.6-264.4) and 33.5 (0-100.9) thousand under SSP2-4.5, SSP3-7.0 and SSP5-8.5, respectively. As for the SSP1-2.6 scenarios, DAPP was projected to decrease by only 25.1 (0-42.3) thousand.

There were substantial spatial differences in the projected trends in DAPP in China (Fig. 2). To illustrate, we describe the two most extreme scenarios—the *sustainability* scenario (SSP1-2.6) with the greatest decline in DAPP and the *regional rivalry* scenario (SSP3-7.0) with the greatest growth (see the full results in Supplementary Table S1 and Fig. S1). In terms of total change in DAPP, the most increase (56.9%, 38.9%-81.6%) in DAPP from 2019 to 2035 was projected to occur in Northeast China. Northwest China showed the least increase (0.1%, −11.7%-9.4%). Across regions, Henan and Hebei were projected to have

the largest decrease in DAPP. In the *sustainability* scenario (SSP1-2.6), the decrease in DAPP in both provinces was >4 times the national average, and under the *regional rivalry* scenario (SSP3-7.0) scenario, DAPP was >2 times the national average. Considering the changes in DAPP per capita in 2019–2035 (Figs. 2c, d), Northeast China was the region expected to witness the most increase (51.8%, 37.1%-73.8%).

### Factors contributing to DAPP

Using the decomposition method[7,14] (see Materials and methods for details), we compared the effects of the four factors that contribute to the change of DAPP. Figure 3 shows the contribution of the four factors (i.e., age structure, total population, air quality, and disease mortality) on DAPP between 2019 and 2025, and between 2025 and 2035, respectively. Among the four factors, population aging was the only factor contributing to the growth of DAPP in the *sustainability* (SSP1-2.6) and *fossil-fueled development* (SSP5-8.5) scenarios, resulting in increases of 641.0 thousand and 668.6 thousand deaths, respectively. In

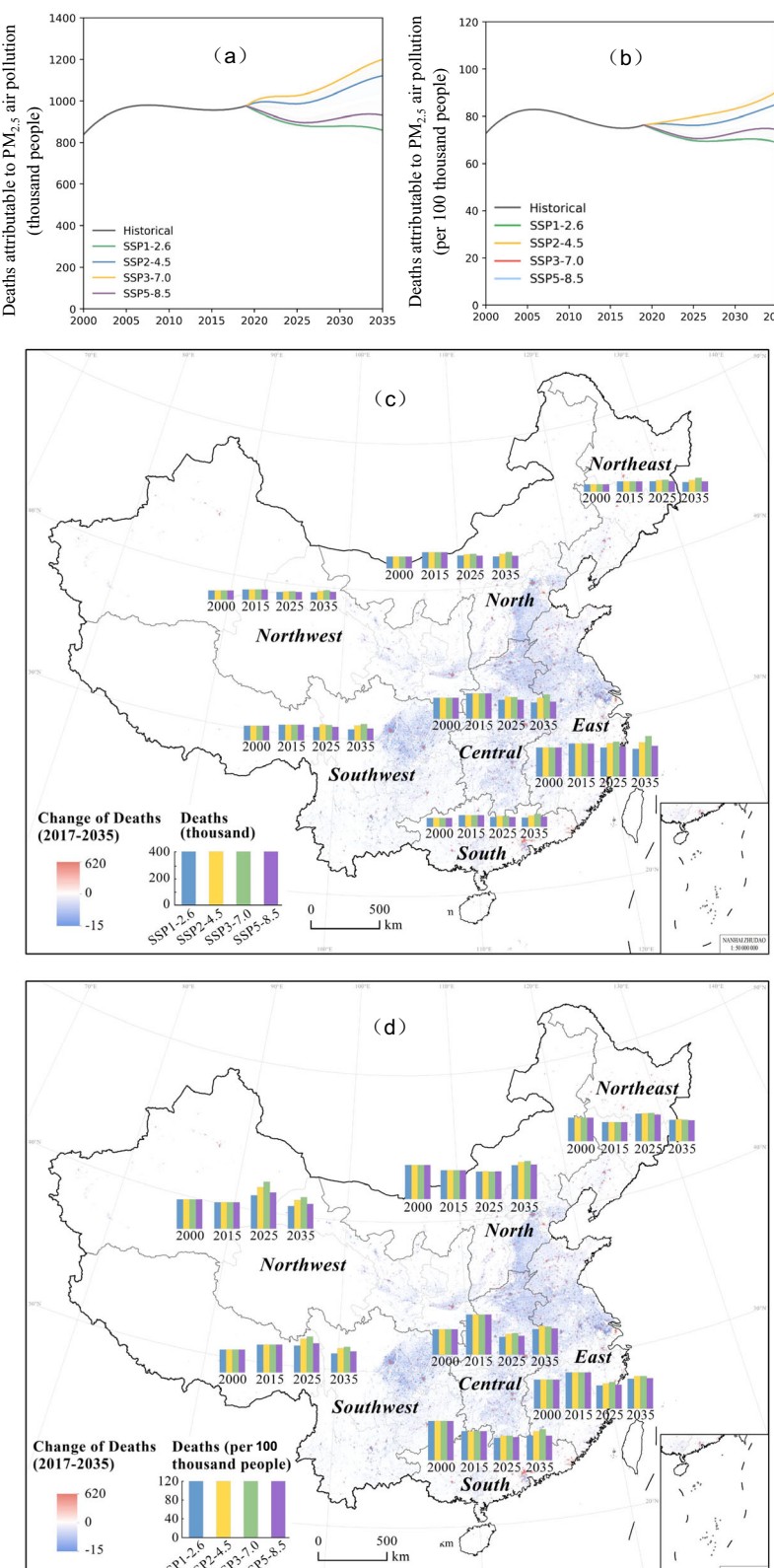

**Fig. 2 | Varying trends in deaths attributable to PM$_{2.5}$ air pollution (DAPP) from 2000 to 2035.** Note: changes in the total amount of DAPP (**a**) in China and among regions (**c**), and changes in DAPP per capita (**b**) in China and among regions (**d**). The values for each region are the average estimates among the provinces in this region (see Supplementary Table S2 for values).

the *middle of the road* (SSP2-4.5) and *regional rivalry* (SSP3-7.0) scenarios, although population growth and the pollution also slightly increased growth in DAPP, >90% of growth was due to population aging. Specifically, in these two scenarios, population aging increased DAPP by

728.7 thousand and 632.3 thousand, respectively. Reduction in disease mortality was responsible for >69% of the reduction in DAPP, decreasing DAPP by 551.1 thousand, 522.8 thousand, 509.0 thousand, and 562.7 thousand from 2019−2035 in the four scenarios, respectively.

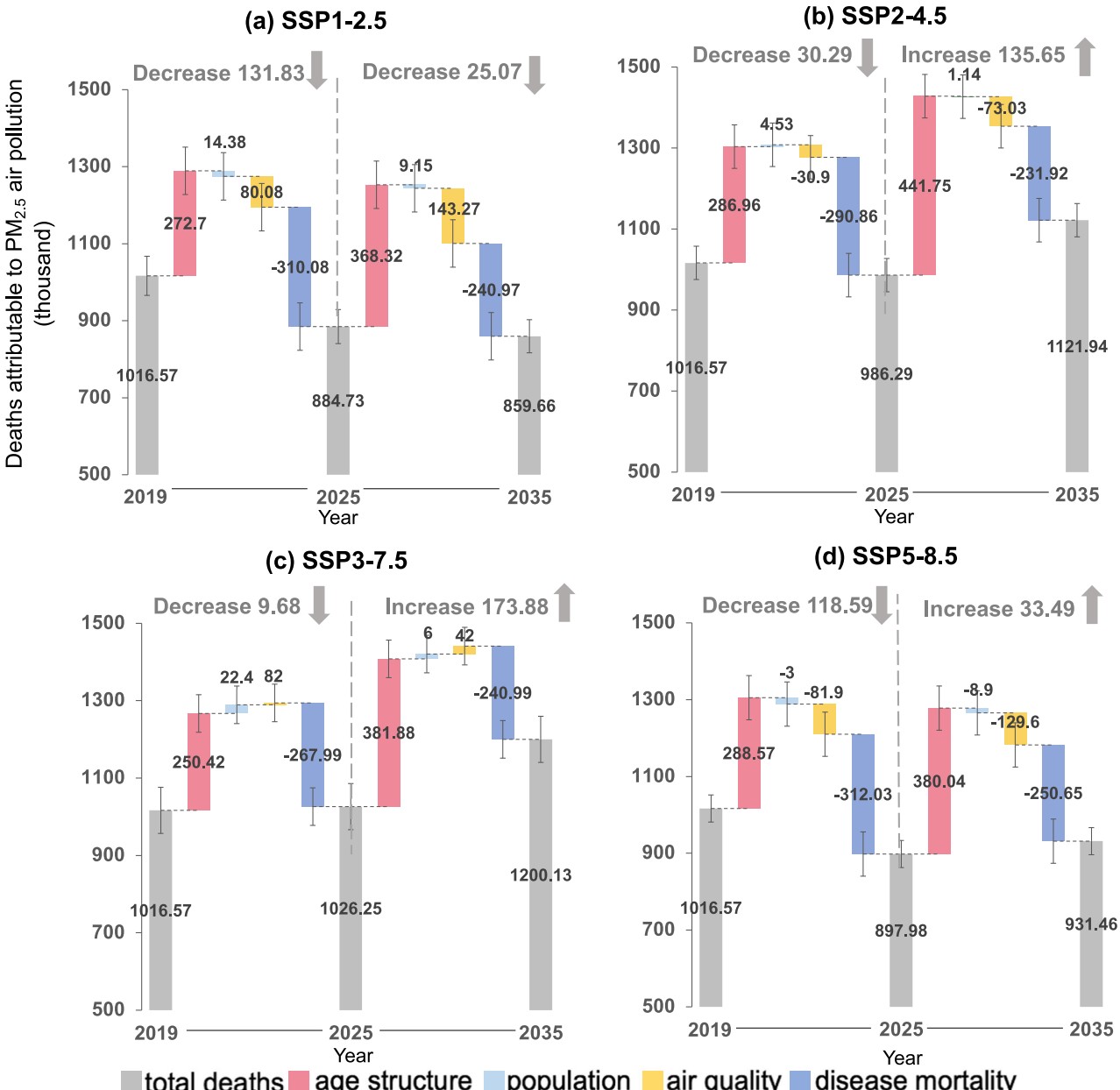

**Fig. 3 | Contributions of different factors to deaths attributable to PM$_{2.5}$ air pollution (DAPP) changes in China from 2019 to 2035.** Note, these plots show the cumulative effect of four factors: age structure, total population, air quality, and disease mortality, on the DAPP between 2019–2025 and 2025–2035 in SSP1-2.6 (**a**), SSP2-4.5 (**b**), SSP3-7.0 (**c**), SSP5-8.5 (**d**) scenarios. Error bars represent 95% confidence intervals.

From 2025 to 2035, the age structure was the most important factor driving the increase in DAPP. The increase in the older population has led to an increase of 279.8 (250.4-288.6) thousand DAPP from 2019 to 2025. The impact of disease mortality on DAPP was projected to level off. From 2025 to 2035, the reduction in disease mortality reduced DAPP by 241.0 (231.9-250.7) thousand, which is 59.5 thousand fewer than the value during 2019–2025. The impact of changing age structure on DAPP began to exceed the impact of declining disease mortality on DAPP and became the main reason for the increase in DAPP from 2025–2035.

The contributions of different drivers to DAPP showed certain differences at the provincial scale (Supplementary Fig. S5). For regions with higher rates of aging, such as Heilongjiang, Jilin and Liaoning, age structure was the most important factor affecting the change in DAPP. Affected by the aging of the population, the total DAPP in these regions continued to rise under all four scenarios, and the total increase was

about 1.2 times the national average. Taking Liaoning Province with the highest level of aging population as an example, in the four scenarios, the total DAPP increased 1.9 and 4.8 thousand in SSP1-2.6 and SSP5-8.5 respectively. And its total DAPP increased by 12.3 and 4.8 thousand in SSP2-4.5 and SSP3-7.0, respectively, which will be 2.5 times and 2.8 times higher than the average change in the national DAPP. In Tibet, Ningxia, and Guizhou, where the proportion of older adults is lower, the growth in DAPP caused by population aging was projected to be less than half of the average change in the national DAPP. Although the reduction in DAPP caused by factors such as the improvement of medical conditions in these regions is also lower than the average of all provinces in the country, overall, the DAPP in these regions still shows a downward trend.

In addition, regarding the types of diseases and considering all age groups affected by DAPP, the older individuals are still the most susceptible group (Fig. 4 and Fig. S6). By 2035, in any scenario, the

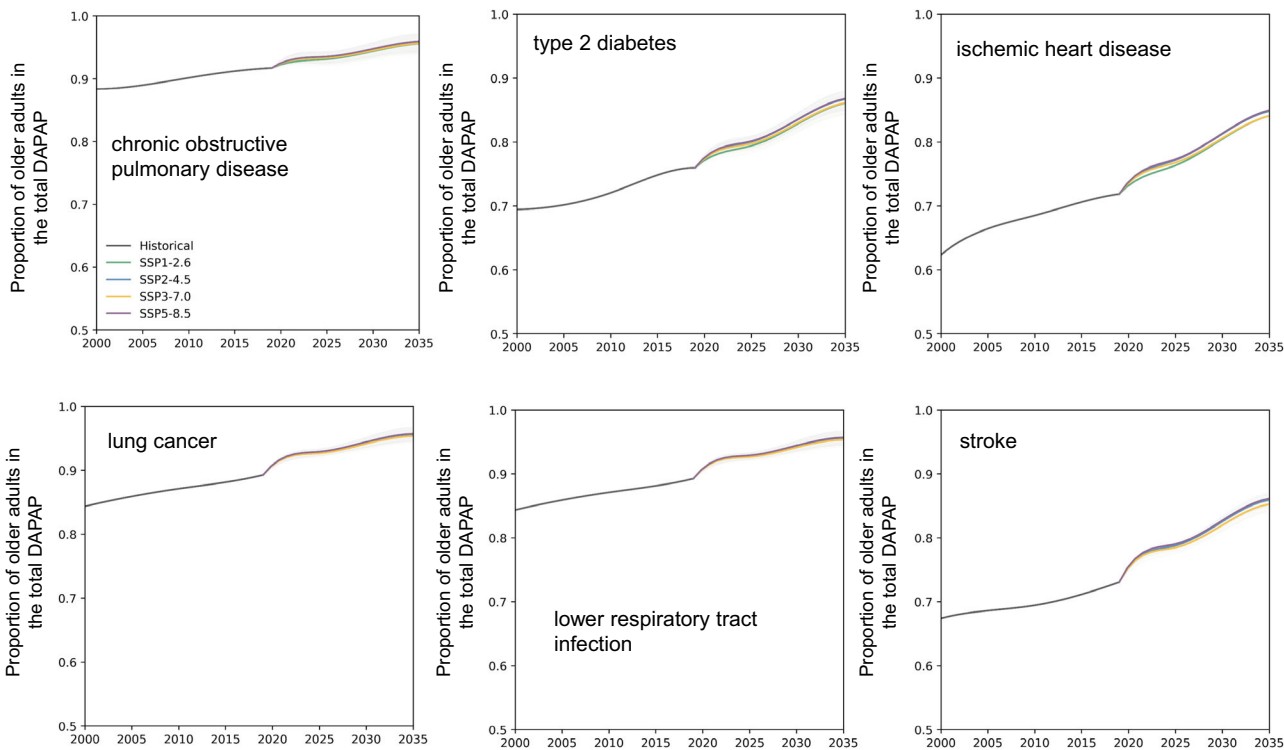

**Fig. 4 | The proportion of deaths among the older population by disease from 2000 to 2035.** Data are presented as mean values, and shading indicates the 95% confidence interval.

average proportion of older adults dying from six diseases (i.e., chronic obstructive pulmonary disease, lung cancer, lower respiratory infection, ischemic heart disease, stroke and type 2 diabetes) will account for >79% of the total DAPP. Specifically, the percentage of these individuals with chronic obstructive pulmonary disease and lower respiratory infection will be >95% and 93%, respectively. Furthermore, the percentage of DAPP in older individuals for all six diseases will rise steadily from 2019 to 2035. Among them, the percentage attributable to lung cancer will increase the most, growing from 66.3% in 2019 to 79.7% in 2035 under SSP1-2.6, or showing an average increase of 13.8% in the four scenarios.

## Discussion
### Population aging is the most important challenge
Population aging is the leading challenge China faces in dealing with DAPP in the future. Our study found that population aging will be the primary contributor to increases in DAPP from 2019–2035 (Fig. 5). First, although the proportion of older adults in the population is relatively small, older adults will account for an outsized fraction of total DAPP. Between 2019 and 2035, while the proportion of older adults will range from 10% to 35% in most regions, >90% (e.g., almost 100% in Northeast China) of the growth in DAPP will be due to population aging. These estimates are also in line with previous studies. For example, Bu et al.[1] found that older adults have been disproportionately affected by DAPP from 1990–2019 on a global scale, and Wang et al.[23] attributed DAPP growth in China by 2020 and 2030 largely to population aging. Second, population aging will be the only factor increasing DAPP for most scenarios, except for the *regional rivalry* scenario (SSP3-7.5), in which population growth and air pollution will also make a small contribution (-2% only) to the increase in DAPP.

The major DAPP-related challenges posed by population aging are mainly demonstrated in the following four aspects. First, the adverse impact of air pollution on people's health will become more severe with age. Since human experience weakening physiological and metabolic processes when getting older[10] and people are exposed to

poisonous substances in ambient conditions that are absorbed by and then accumulate in the human body[27], older adults are the most vulnerable to air pollution. Our findings show that by 2035, the average proportion of older adults dying of diabetes mellitus type 2 and ischemic heart diseases will account for over 85% of the total deaths of all ages. Second, among the six diseases related to DAPP, the proportions of deaths of older adults will consistently increase, and three types of diseases (i.e., lung cancer, ischemic heart disease and stroke) will increase by over 12%. Third, it is projected[28] that by 2022 China will enter a stage where older adults will account for >14% of the total population. By 2035, the proportion of individuals aged 65 years or older will account for 20% of the total population, further intensifying the situation. The rate of population aging in China has exceeded that in most developed countries[29]. As population aging continues, DAPP is likely to subsequently increase[30]. Fourth, global climate change may have a compounding effect, exacerbating the vulnerability of older adults. The increased frequency and intensity of extreme events such as heat waves, floods, droughts and wildfires under climate change (especially under the SSP3-7.0 scenario with high population growth and high emissions)[31] may also increase the morbidity among older adults[32]. In increasingly extreme climates, older adults are more physically and mentally vulnerable and at greater risk of premature death. For example, Yuan et al. showed that, for every 1 °C increase in temperature, the health score of older workers (aged 60 and above) decreases by about 6.4 times as much as that of young and middle-aged workers (aged 15–59)[33].

In addition to directly causing the DAPP, the aging of the population may also affect DAPP through their energy consumption behavior and household consumption structure, and corresponding air pollutant emissions. Existing knowledge on the associations between population ageing and air pollutants are mixed and piecemeal. For example, some studies found that older adults may consume more fossil energy[21,34] and energy intensive products and services[35], while others found that older adults may consume less fossil energy[36]. In the context of an aging society[37], how different age groups' energy

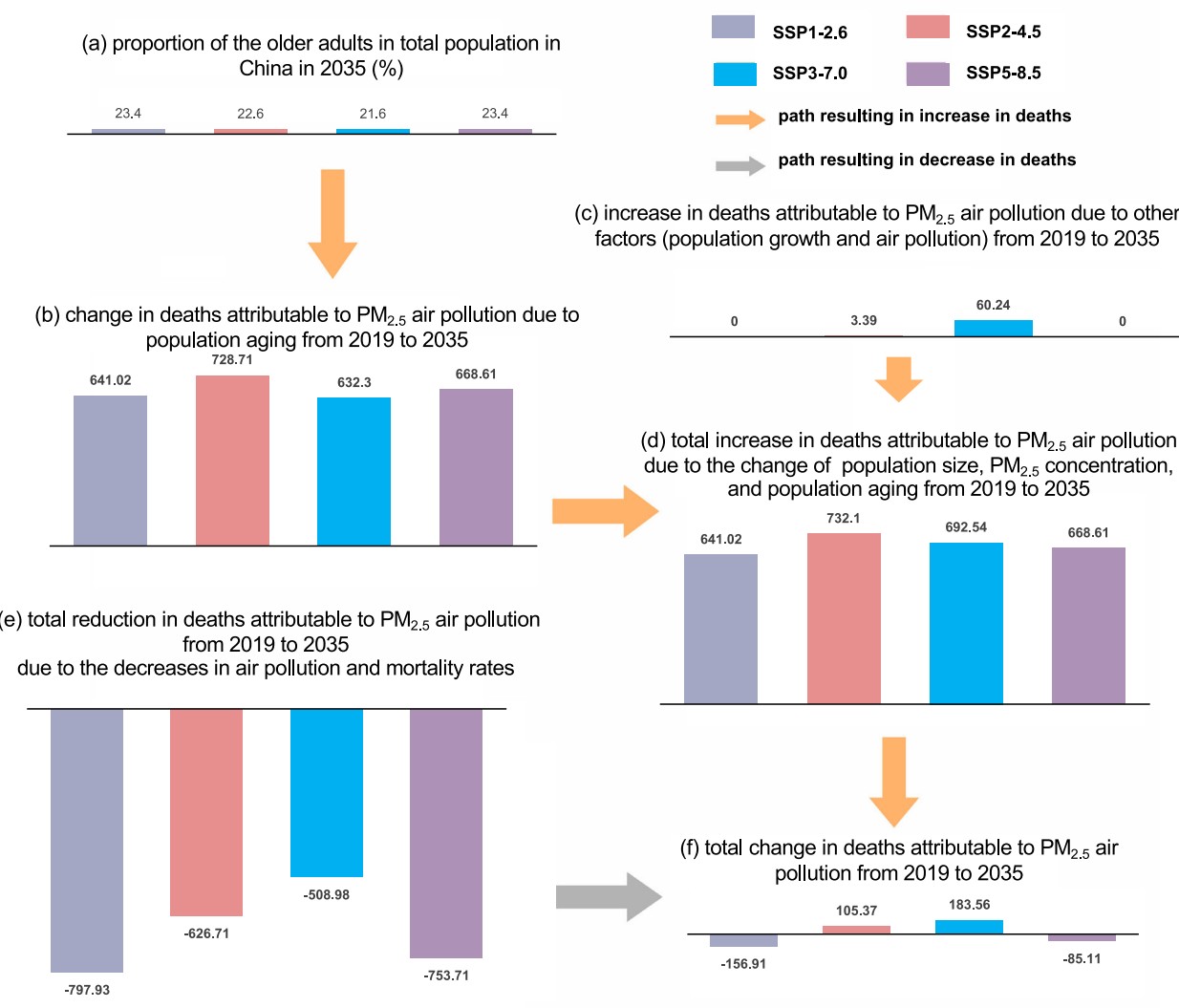

**Fig. 5 | Contribution of age structure change to deaths attributable to PM$_{2.5}$ air pollution in four scenarios: SSP1-2.6, SSP2-4.5, SSP3-7.0 and SSP5-8.5. a–d** depict the paths leading to increase in the deaths; **e** is the path resulting in decrease in the deaths; and **f** represents the net changes in the deaths.

consumption behaviors and household consumption structure affect air pollution needs further investigation.

Hence, to alleviate the health burden caused by air pollution and decrease health risks, apart from improving air quality, China should pay special attention to older adults who are directly exposed to air pollution. First, formulating plans for promoting public health awareness is suggested. This will encourage older adults to conduct indoor activities during pollution events, thus reducing direct exposure to polluted air. For example, the U.S.A. and European Union countries have implemented public health awareness plans[10] to provide information on air pollution for older adults, encouraging them to reduce their activities in highly contaminated conditions. However, indoor air pollution also needs to be considered, as the average adult in China spends around 86% of their time indoors[38]. For example, Yun et al. pointed out that indoor pollution caused by solid fuel use is far more serious than other activities, and promoting clean cooking and heating is an effective way to improve indoor air quality[37,38]. Second, the government should improve health care by supporting additional investments in health care and adopting targeted measures to improve older adults' health care insurance and alleviate their economic burden[27]. Other policies, such as providing people residing in polluted areas with

medical compensation, incorporating medicines targeted at specific diseases into medical insurance and reducing the medical cost of treating air pollution-related diseases, are suggested as well.

### Insights on prevention

The difficulty in preventing and mitigating DAPP has regional differences in China. Our results showed that, while Eastern, Southeast, and Southwestern China have made some positive progress in preventing air pollution and improving health care[39,40], there is still room for improvement in Central, West, and Northeast China which have a rapidly aging society[25]. Such regional differences also echo the challenges faced by countries with similar levels of development (Table 1).

First, air pollution poses a huge threat to health in less economically developed regions, such as Central and West China. This is also true globally. In the assessment of air pollution and health from the WHO[41], nearly 90% of air pollution-related deaths occurred in low-income and middle-income countries (e.g., South Asian and Western Pacific countries). Due to limitations in technology, most developing countries experience substantial air pollution[42]. Lessons should be drawn from developed countries in terms of adjusting their industrial

**Table 1 | Health risks of air pollution in countries with varied incomes**

| World (2015) | | | China (2015) | | |
|---|---|---|---|---|---|
| Income levels | Percentage of older adults (65 years or more) (%) | Death rate attributable to air pollution (per 100 thousand) | Regions | Percentage of older adults (65 years or more) (%) | Death rate attributable to air pollution (per 100 thousand) |
| High income | 22.5 | 0.7 | East and South | 22.2 | 1.2 |
| Upper-middle income | 12.6 | 0.9 | North and Northeast | 28.4 | 1.5 |
| Lower-middle and low income | 4.8 | 1.3 | Central and Northwest | 21.7 | 1.7 |

Global population data and classifications by income levels were derived from publicly available World Bank information. Death rate data were accessed from Cohen et al.[14]. The Chinese classifications by income levels are based on income rankings of China.

structure and exploiting clean energy to reduce air pollution, as well as improving healthcare.

Second, for regions that have achieved greater economic development (e.g., Northeast China), the threat of population aging was the main driver of increased DAPP. For these regions, providing healthcare and improving basic pension plans for older adults is a priority, for instance, establishing long-term nursing systems to meet the requirements of older adults[43]. We also note that the air pollutants generated in Northeast China may lead to haze episodes in neighboring provinces[6] (e.g., Beijing-Tianjin-Hebei urban agglomeration) via atmospheric pollution transport. These spatial spillover effects require collaboration among regions to mitigate the adverse effect of air pollution.

Third, for regions with higher levels of economic development (e.g., East China), the decrease in DAPP brought by declines in the death rates of diseases and improved air quality can effectively offset the increase in DAPP led by population aging, due to reduced emissions of atmospheric particulates and improvements in health care facilities[44]. Their successful experiences should be translated to other regions. For example, U.S.A.[45] and the European Union[46,47] continuously optimized the health care system and improved air quality, which were effective for controlling DAPP. Meanwhile, European countries have jointly developed emissions reduction technology and encourage green business to reduce industrial pollution, thus reducing health risks[46]. It is also worth mentioning that some developed regions improved air quality by outsourcing heavily polluted manufacturing to developing regions[48]. Therefore, regional collaboration is encouraged to jointly prevent pollution leakage and spillover effects.

To improve the health and well-being of residents, relevant policies should consider the aging population. At the government level, the consideration of aging needs to be fully integrated into health policies to build an age-friendly society. In addition, it is imperative to increase investment in healthcare to reduce the economic burden of older adults.

## Limitations of the study
There are several sources of uncertainty and limitations in our research. First, we adopted the MR-BRT model developed by the World Health Organization (WHO) and Global Burden of Disease (GBD). To assess the impact of this choice we compared the results of this study with deaths calculated by Geng et al.[49] based on the Global Exposure Mortality Model (GEMM), which models all-natural cause mortality and does not incorporate additional types of exposure, such as active smoking[50]. Although the absolute values of the two results differ, the trends are consistent. Second, we assumed a constant association estimate for population vulnerability to $PM_{2.5}$ air pollution. The combined effects of air pollution and other risk factors (such as extreme temperature) on some diseases and the change in vulnerability to air pollution in our study may be conservative[51–53]. Third, when estimating the health burden of air pollution, we considered both the effects of climate change and socioeconomic factors, but there are still some other factors that remain difficult to take into account. For example,

we did not consider future policy interventions or indoor air pollution. Future public emergencies (e.g., pandemic) or possible policies (e.g., emissions reduction measures and temporary lockdowns) will affect the exposure of air pollution. Fourth, the population forecasts used in this study are based on assumptions on all-cause mortality that are not consistent with those from which the cause specific mortality rates have been derived in the GBD, which might also influence our estimates in DAPP. Therefore, the projections in this study only reflect long-term trends in the health effects of outdoor $PM_{2.5}$ air pollution.

## Methods
### Study area and scenario settings
This paper focuses on the change in key factors contributing to DAPP in China at the provincial scale. The Global Burden of Disease study showed that $PM_{2.5}$ pollution caused 4.2 million deaths worldwide in 2015, of which China was the country with the largest number of deaths, reaching 1.1 million (IHME and HEI, 2017). In addition, China is the country with the largest population of older adults in the world[11]. By the end of 2019, the population aged over 65 years in China was 173 million, accounting for 14% of the world's older population[54].

This study assessed 31 provinces, autonomous regions, and municipalities, with Hong Kong, Macau and Taiwan excluded due to data limitations. The 31 provinces were aggregated into North, Northeast, East, Central, South, Southwest, and Northwest regions (Supplementary Fig. S2). From 2000 to 2019, the population-weighted $PM_{2.5}$ concentration and population aging in China showed an upward trend. The average population-weighted atmospheric $PM_{2.5}$ concentration in China increased from 28.1 ug/cm³ to 40.4 ug/cm³, an increase of 43.9% (Global $PM_{2.5}$ Assessment Dataset[55]).

ScenarioMIP classifies the scenarios of future socioeconomic and climate change into eight primary and secondary scenarios based on their priorities[18,56]. In this study, we selected four scenarios with high priority in the first-level experiment to estimate DAPP. Among them, SSP1-2.6 is a sustainable green path characterized by low vulnerability and low mitigation challenges, with a mild climatic change and a small range of variation[18,24]. In this scenario, the birth rate, death rate and migration rate are lower; population is more educated; and population is older[15]. SSP2-4.5 is an intermediate path with a moderate climatic change characterized by moderate social vulnerability and a moderate climatic change. In this scenario, social, economic and technological trends are similar to historical ones[15], so that the changes in fertility, mortality, migration and the proportion of the older population also maintain current trends. SSP3-7.0 is a combined scenario of regional competing paths characterized by higher social vulnerability and higher mitigation challenges with a moderate to severe climatic change. In this scenario, population grows rapidly; the regional development is unbalanced, and the per capita economic and technological development level is low[15]. With rapid growth of population size, the fertility rate and mortality rate are relatively high, and the degree of population aging is relatively low[24]. SSP5-8.5 is a scenario dominated by traditional fossil fuel combustion, characterized by high social vulnerability and high mitigation challenges, with a severe

**Table 2 | Data sources**

| Data | | Resource |
|---|---|---|
| Historical data (2000-2019) | Concentration of PM$_{2.5}$ on surface (spatial resolution 0.1°); | Global PM$_{2.5}$ Assessment Dataset[55] |
| | Disease mortality data (provincial scale); | Yin et al.[25] |
| | Disease mortality data (national scale) | GBD2019[64] |
| | Population, GDP per capita, proportion of secondary industry, educational level, fertility rate (provincial scale). | China Statistical Yearbooks[54] |
| Projected data (2019–2035) | Sea salt, sulfate, organic aerosol, black carbon (spatial resolution 1°); | CMIP6 dataset[57] |
| | Population datasets in China | Jiang et al.[58], Chen et al.[26] |
| Other data | Provincial administrative boundary | 1:1 million released by the National Bureau of Surveying, Mapping and Geographic Information |

climatic change. In this scenario, social development will emphasize economic growth, while the fertility rate, mortality rate and migration rate are relatively low; and the degree of population aging is relatively high[24].

## Data

The data used in this study includes pollutant concentration data, disease mortality data, historical socioeconomic data, air pollutant composition data, future socioeconomic data, and other geographic data (Table 2).

Future PM$_{2.5}$ composition data came from the CMIP6 dataset. Specifically, the data are based on the emissions, climate and meteorological specification under each scenario in the ScenarioMIP framework and are logically consistent with socioeconomic factors in the SSPs[57]. We used the average of the simulation results of multiple models for each scenario[24].

Future socioeconomic data include data on China's population age structure and population education level at the provincial scale[26] and data on the total population at 0.5° grid cell resolution[58]. Based on the latest provincial and prefecture-level city statistical yearbook data and population data in the RCP pathways, the dataset from Chen et al.[29] estimates the future population of China's provinces under five SSPs from 2010 to 2100. The data reflects the internally consistent relationships and logic of socio-economic development and climate change at the provincial scale. With 2010 as the base year, Jiang et al.[58] accumulated data on key elements such as fertility, mortality, migration rates, population, age structure, and education level at the provincial level on an annual basis out to 2100 under each SSP. Cai et al.[29] considered the birth policy, household registration policy, and population ceiling of megacities implemented in China in recent years with a recursive multidimensional model. They also verified and compared the predicted total population and structured information results based on the latest statistical yearbook data of provinces and prefecture-level cities and the current gridded population data released by international institutions. Based on 2010 statistical data and the third Economic Census, Jiang et al.[58] estimated the labor force level, total factor productivity, and capital stock, and verified the parameters of the economic model. They also estimated China's economic progress under the SSPs using the improved Population, Development and Environment (PDE) model and Cobb-Douglas assumptions.

## Estimating DAPP

First, we used the comparable risk assessment framework[3] to calculate DAPP[51] as a function of the population attributable fraction (PAF), population, disease mortality and age structure[41,59].

$$DAPP = \sum_{a,d} \left( PAF_{a,d} \times POP \times Rate_{a,d} \times AgeP_a \right) \quad (1)$$

where $PAF_{a,d}$ refers to the attributable fraction of disease d in the population of age a, $POP$ refers to the population, $Rate_{a,d}$ refers to the mortality rate of disease d in the people of age $a$, and $AgeP_a$ refers to the proportion of the population of age $a$ in the total population. Among them, age includes 15 stages: 25–30, 30–35,..., 90–95, and >95 years old. Disease includes 6 categories: chronic obstructive pulmonary disease, infectious lower respiratory tract infection, lung cancer, ischemic heart disease, stroke and type 2 diabetes[41]. In addition, $PAF_{a,d}$ is determined by the relative risk $RR_{a,d}$ of death due to disease $d$ in age group $a$[41]. Specifically, it can be expressed as[14],

$$PAF_{a,d} = \frac{RR_{a,d} - 1}{RR_{a,d}} \quad (2)$$

when the PM$_{2.5}$ exposure level is higher, the $RR_{a,d}$ will be higher[3,60]. The quantitative relationship between them can be obtained from the exposure-response function. The exposure-response function used in this study comes from the Bayesian, regularized, trimmed (MR-BRT) model recently announced in GBD2019[3], and the DAPP estimated based on the median value of the MR-BRT model as the primary result[3].

## Projecting DAPP from 2019 to 2035

We calculated future the PM$_{2.5}$ concentration in China based on empirical formulas and future pollutant concentration data. We used the GBD model to estimate future disease mortality in China and quantified DAPP based on a comparable risk assessment framework.

Specifically, referring to the study of Chowdhury et al.[61] and Van Donkelaar et al.[62], we used the empirical formula to estimate PM$_{2.5}$ concentration:

$$PM_{2.5} = BC + OA + SO_4 + NH_4 + 0.25 \times SS + 0.1 \times dust \quad (3)$$

$$NH_4 = \frac{(36 \times SO_4)}{96} \quad (4)$$

where $BC$ represents black carbon, $OA$ represents organic aerosol, and $SS$ represents sea salt. Since the CMIP6 does not report the concentration of surface $NH_4$, we assumed that $NH_4$ exists only in the form of ammonium sulfate, and the concentration of $NH_4$ was estimated from the concentration of $SO_4$.

Since the PM$_{2.5}$ concentration obtained from the simulation is lower than the in situ observations, we calibrated the PM$_{2.5}$ concentration calculated by the model and the observations of the baseline period (2015-2019) as follows:

$$PM_{2.5_f^{cal}} = PM_{2.5_{bl}^{obs}} \times \frac{PM_{2.5_f^{est}}}{PM_{2.5_{bl}^{est}}} \quad (5)$$

where $PM_{2.5_f^{cal}}$ represents the calibrated PM$_{2.5}$ concentration in the future; $PM_{2.5_{bl}^{obs}}$ represents the observed PM$_{2.5}$ concentration during

the baseline period (2012-2019), and $PM_{2.5_{bl}^{est}}$ represents corresponding estimated PM$_{2.5}$ concentration during the baseline period obtained from Eq. 3. $PM_{2.5_f^{est}}$ represents the estimated PM$_{2.5}$ concentration in the future using Eq. 3.

In addition, in the GBD2019, the quantification of DAPP required consideration of six diseases: chronic obstructive pulmonary disease, lower respiratory tract infection, lung cancer, ischemic heart disease, stroke, and type 2 diabetes mellitus. Specifically, according to Foreman et al we forecasted the disease mortality based on the social-ecnomic development level[63]. According to the forecasting model, the following relationship is established between disease mortality and its drivers:

$$\ln(m) \sim N(\hat{y} + \hat{\epsilon}, \sigma) \tag{6}$$

where $m$ represents the mortality rate of a specific disease, whose value is mainly determined by $\hat{y}$ (representing variables such as long-term trends, socioeconomic development, and risk factors) and $\hat{\epsilon}$ (representing residuals that cannot be explained by the above factors).

$$\hat{y} = \beta_1 SDI_{<0.8} + \beta_2 SDI_{\geq 0.8} + \theta_a t + \alpha_{la} \tag{7}$$

$$\hat{\epsilon} = ARIMA\left(\epsilon_{history}\right) \tag{8}$$

where $\beta_1$ and $\beta_2$ represent the global coefficients when the socio-demographic index (SDI) is <0.8 and >0.8, respectively. $\theta_a t$ represents the long-term trend for a specific age group, and $t$ represents time. $\alpha_{la}$ refers to the intercept for a specific age in a specific region. $\hat{\epsilon}$ can be calculated by the autoregressive integrated moving average function.

Since estimating just 6 causes of mortality may vary from the estimates of total mortality from all causes, we constrained our estimates of DAPP from the 6 causes to the all-cause mortality envelope at the national level published by GBD 2019 (Supplementary Fig. S3). First, according to the historical mortality rates of all-cause (level 1 causes) at the national level provided by GBD2019, we estimated the mortality rates of all-cause diseases during 2019–2035 based on Eq. (7). These estimates formed the "envelope" to constrain the level 2 causes (Supplementary Fig. S3). Second, we estimated the mortality rates of diseases at level 2, i.e., non-communicable diseases (NCD), communicable, maternal, neonatal and nutritional disease (CMNND) and injuries during 2019–2035 based on Eq. (7), according to historical data provided by GBD2019. We then constrained the mortality rates of the three level 2 diseases to the "envelope" of all causes (level 1 causes) as follows,

$$Rate_{1stconstrained,a,y} = \frac{Rate_{NCD,y} + Rate_{CMNND,y} + Rate_{injuries,y}}{Rate_{all,y}} \times Rate_{a,y} \tag{9}$$

where $Rate_{1stconstrained,a,y}$ is the constrained mortality rate for the disease $a$ in year $y$. $Rate_{NCD,y}$, $Rate_{CMNND,y}$, $Rate_{injuries,y}$, and $Rate_{all,y}$ are the original mortalities rates estimated for NCD, CMNND, injuries and all causes based on Eq. (7), respectively. $Rate_{a,y}$ is the original mortality rate of disease $a$ in year $y$ calculated by Eq. (7). Third, we further constrained the mortality rates of the six diseases at level 3 to the envelope of the constrained level 2 mortality rates above in a similar way. Taking NCDs as an example, we constrained the mortalities rates of the five level 3 causes (COPD, DM2, IHD, LC, and stroke) and the mortality rates of the other causes (rest-NCD for short) to the constrained level 2 mortality rates (i.e., $Rate_{1stconstrained,a,y}$) as below,

$$Rate_{2ndconstrained,n} = \frac{\sum Rate_{n,y} + Rate_{rest-NCD,y}}{Rate_{1stconstrained,NCD,y}} \times Rate_{n,y} \tag{10}$$

where $Rate_{2ndconstrained,n}$ is the constrained mortality rate of a level 3 disease $n$ in year $y$. $Rate_{n,y}$ represents the original mortality rate of a level 3 disease $n$ in year $y$ estimated by Eq. (7). $Rate_{n,y}$, and $Rate_{rest-NCD,y}$ are the mortalities rates of a level 3 NCD estimated by Eq. (7) using historical data from Yin et al.[25], and the mortality rate of rest NCDs estimated by Eq. (7) based upon GBD2019, respectively. The constrained mortality rate of LRI was calculated in a similar way. After two rounds of the constraining process, our estimates for the 6 causes of mortality are within the range of all-cause mortality, and are consistent with GBD2019.

Finally, based on the estimated results of PM$_{2.5}$ concentration and disease mortality as well as the population and age structure data, we quantified the DAPP from 2020 to 2035 with the comparable risk assessment framework model (i.e., Eq. (1)). In addition, the upper and lower bounds of the confidence interval for DAPP in this study were determined by the upper and lower bounds of the exposure response function and the upper and lower bounds of confidence intervals for PM$_{2.5}$ concentration and disease mortality. Among them, the upper and lower limits of the 95% confidence interval range of the exposure-response function are the 2.5% and 97.5% quantiles of each result in the 1000 curve fits of the MR-BRT model[3]. The confidence interval of the PM$_{2.5}$ concentration was determined by the difference of 11 models in the CMIP6 dataset[61]. This study assumed that the simulation results of each model conformed to the t-distribution[57], and the 95% confidence interval of the PM$_{2.5}$ concentration can be expressed as:

$$C_{CI} = Con_{mean} \pm \sqrt{Var(C)} * T_{95\%}(n-1) \tag{11}$$

where $C_{CI}$ represents the uncertainty interval of the PM$_{2.5}$ concentration; $C_{mean}$ refers to the mean value of the PM$_{2.5}$ concentration obtained by multiple models; and $Var(C)$ refers to the variance in the PM$_{2.5}$ concentration obtained from multiple models. $T_{95\%}(n-1)$ represents the t-statistic with $n$-1 degrees of freedom under the 95% confidence interval, where $n$ is the number of correlated modes.

Disease mortality is determined by $\epsilon$ in Eq. (8)[63], $\epsilon$ represents the residual between the estimated value and the actual value, obeying the mean of 0, and the variance of $Var(e_0)$ is normally distributed. Therefore, the 95% confidence interval for $ln(m)$ can be expressed as:

$$\ln(m)_{CI} = \hat{y} + \hat{\epsilon} \pm \sqrt{Var(\epsilon - \hat{\epsilon})} * T_{95\%}(n-2) \tag{12}$$

where $ln(m)_{CI}$ represents the uncertainty interval for the natural logarithm of mortality from a specific type and age group of diseases; the sum of $\hat{y}$ and $\hat{\epsilon}$ represents the predicted value of $ln(m)$, and $Var(\epsilon - \hat{\epsilon})$ refers to the variance of the residual term in the regression model (Eq. 10). $T_{95\%}(n-2)$ represents the t-statistic with n-2 degrees of freedom under the 95% confidence interval, where n is the sample size for constructing the regression model. $Var(\epsilon - \hat{\epsilon})$ can be further obtained by the following formula.

$$Var(\epsilon - \hat{\epsilon}) = \frac{\sum e_i^2}{n-2} \times \left[\frac{1}{n} \times \frac{\left(x_f - \bar{x}\right)^2}{\left(x_i - \bar{x}\right)^2}\right] \tag{13}$$

where $\sum e_i^2$ is the residual sum of squares of the regression model, $x_i$ and $\bar{x}$ represent the mean of each independent variable and independent variable in the regression model, respectively, and $x_f$ represents the input value of the independent variable when predicting the future based on the regression model. We reported the 5-year intervals to smooth out the influence of abrupt changes due to data artifacts (Fig. 1 and Fig. 2).

We further compared DAPP calculated for historical periods with calculated by previous studies and found that previous studies[7,25,49] showed similar results to ours. Although some estimates based on the Global Exposure Mortality Model (GEMM) model were higher than our

estimates[49], the results showed a similar trend over time. This is because the relative risk estimated by GEMM model are higher compared to the relative risk values obtained by the Bayesian, regularized, trimmed model, especially in areas with high $PM_{2.5}$ concentrations[9]. Also, we compared the results estimated under different future scenarios with the results estimated by Global Burden of Disease−Major Air Pollution Sources (GDBMAPS)[2] (Supplementary Table S3). Different from the four scenarios in this study, the results in GBDMAPS were mainly based on different emission pathways under different energy utilization and air pollution control policies. Consistent with our findings, DAPP were lower under the sustainable development paths with policy interventions (such as the BAU2 and PC2 scenarios in GDBMAPS China[2] and the SSP1-2.6 scenarios in this study). However, since we only considered outdoor air pollution when quantifying DAPP, ignoring indoor pollution and its interaction with outdoor air pollution, the amount of DAPP in this study is lower than the results of GBDMAPS China. In addition, the disease mortality data used in GBDMAPS was simulated based on the estimated value of GBD2013. It was assumed that the change trend of disease mortality is consistent with the change trend from 1990 to 2013, and the changes in socio-economic levels such as medical conditions across different scenarios were not considered. Therefore, the disease mortality in GDBMAPS China were higher than the ones in our study. This is another reason for the higher estimated DAPP estimated by GBDMAPS than our estimates. We used Pearson's correlation to verify the reliability of the results (Supplementary Fig. S3). Specifically, we compared the results obtained in Eq. (1) with the estimates calculated by Yue et al.[7] based on the Integrated Exposure Response (IER) model, the estimates calculated by Geng et al.[49] based on the Global Exposure Mortality Model (GEMM), and the estimates calculated Yin et al.[25] at the provincial scale in China.

### Decomposing the contributions to DAPP

In addition to being affected by the age structure of the population, DAPP is also affected by the effects of population size, $PM_{2.5}$ concentration and related disease mortality. Specifically, the total population and proportion of aged population, $PM_{2.5}$ concentration, and disease mortality are all positively correlated with DAPP. Based on Cohen et al.[14] and Yue et al.[7], we quantified the influence of age structure and population size, $PM_{2.5}$ concentration and related disease mortality on DAPP using the decomposition method. The advantage of decomposition method is to transform the nonlinear relationship between individual drivers and DAPP changes into a simple linear relationship[14]. Compared with simple sensitivity analysis, decomposition method considers the interaction of various factors, which makes the number of DAPP correspond to the contribution of driving factors. The net change of DAPP in a certain period can be expressed as the sum of the contributions of various driving factors to the change of DAPP, which is helpful for the quantitative analysis of the driving mechanism of DAPP. At the same time, the decomposition method converts the changes of different driving factors into contributions to the changes of DAPP, which supports the judgment of the relative importance of each factor.

In the decomposition analysis (Supplementary Fig. S4), we gradually introduced the changes in age structure, population, $PM_{2.5}$ concentration and disease mortality separately to calculate DAPP. The difference between the pre-introduction and post-introduction results is the contribution of age structure. In addition, considering that the order of introduction can affect the calculation results, this study calculated all 24 possible combinations of the order of the introduction and used the average of the contributions of the four drivers in each case as the final result. Based on the decomposition analysis, we calculated the driving factors of DAPP in China in 2019–2025, 2025–2030, and 2030–2035 and then analyzed the synergistic driving effect of age structure and other factors on DAPP.

## Reporting summary

Further information on research design is available in the Nature Portfolio Reporting Summary linked to this article.

## Data availability

All the data created in this study are openly available at Github repositories with the identifier https://github.com/q22huang/DAPAP. The source data underlying Figs. 1–5 are provided as a Source Data file. Historical data of concentration of $PM_{2.5}$ on surface from Global $PM_{2.5}$ Assessment Dataset can be obtained at http://fizz.phys.dal.ca/~atmos/. The disease mortality data (national scale) from GBD2019 can be obtained at https://vizhub.healthdata.org/gbd-results/. The historical data of population, GDP per capita, proportion of secondary industry, educational level, fertility rate (provincial scale) can be obtained from China Statistical Yearbooks. The projected data of Sea salt, sulfate, organic aerosol, black carbon (spatial resolution 1°) from CMIP6 dataset can be obtained at https://pcmdi.llnl.gov/CMIP6/. The projected data of population datasets in China from Chen et al. can be obtained at https://doi.org/10.6084/m9.figshare.c.4605713.v1. Other data are available from the corresponding author upon request. Source data are provided with this paper.

## Code availability

The detailed equations for the integrated exposure-response function is accessible to all users at http://ghdx.healthdata.org/record/ihme-data/gbd-2019-burden-risk-1990-2019. Custom Python scripts for estimating disease mortality and the functions used for estimating the deaths attributable to PM2.5 air pollution embedded in the Microsoft Excel file were available at Zenodo (https://doi.org/10.5281/zenodo.8128795).

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

## Acknowledgements

We thank Prof. Aaron J. Cohen (Health Effects Institute, Boston, USA) for his insightful comments on the manuscript. This work was supported by the National Key Research and Development Program of China (Grant No. 2019YFA0607203, QH and CH), Beijing Nova Program (Grant No. 20220484163 QH), the National Natural Science Foundation of China (Grant Nos. 41971225, QH & 41971270, CH) and Natural Science Foundation of Shandong Province (ZR2022QD051, HY).

## Author contributions

F.X., Q.H., H.Y., and C.H. designed the study and planned the analysis. F.X., and H.Y. prepared the basic data. F.X., Q.H., and B.A.B. did the data analysis. F.X., and Q.H. drafted the manuscript. X.F., H.X., P.Y., and B.A.B. contributed to the interpretation of findings, provided revisions to the manuscript.

## Competing interests

The authors declare no competing interests.
