## [Peer Review File · Nature Communications]

nature portfolio

Peer Review FileReviewer comments, first round

Reviewer #1 (Remarks to the Author):

Thank you for giving me the opportunity to review this manuscript. It seems to be a well designed and well performed piece of research on an important topic. The methods appear to be sound and I would recommend publication with relatively minor changes/clarifications.

Specific comments

The title of the paper is slightly confusing. How about "The challenge of population aging for future mitigation of deaths attributable to PM2.5 air pollution in China"?

Minor point: I think it would be better to use "air pollution" rather than just "pollution" throughout (e.g. in the title).

Minor point: I'm not keen on creating new acronyms, such as DAPP. As far as I'm aware, this isn't a common term used in the air pollution field (other than in past papers by the authors).

Abstract

It would be useful to briefly state what the scenarios represent. You might just say e.g. "scenarios representing different levels of sustainable development".

The abstract is a little confusing because it first says "The results show that from 2017 to 2035, DAPP in China will decrease...", then the next sentence says "We found that population aging will be the leading driver contributing to the increase in DAPP...". Should the second sentence say something like "We found that population aging will be the leading driver tending to increase DAPP...".

Introduction

Lines 47-51: Are these lines referring specifically to China? Or global?

Lines 92-95: It might be worth emphasising here that the RCPs represent greenhouse gas concentration trajectories (as opposed to air pollutants).

Line 111: Perhaps clearer to say "...impact of population aging on DAPP in China...".

Results

Lines 161-163: I don't understand the sentence starting "In addition, it is worth noting...". Please can the authors clarify?

Lines 182-185: I also find the sentence here confusing: "In the sustainability scenario (SSP1-2.6), ...".

Lines 187-188: Like the abstract, this sentence is a little confusing because it suggests DAPP increases from 2017 to 2035.

Figure 3: It would be helpful to the reader to explain briefly how to read/interpret the plots (either in the figure title or in the main text).

Discussion

Line 259: I don't think the paper by Wang et al was mentioned in the Introduction when making the case for the research. Can the authors explain how their study compares (and improves upon)

the Wang et al study?

Lines 272-286: The start of the paragraph suggests that "two aspects" will be discussed but the paragraph seems to make three distinct points ("Third, with the fast-growing aging population in China, ...").

Lines 289-292: The KAYA model needs a little further brief description and a reference. What does it do?

Lines 318-339: I'm not entirely sure what point is being made in this paragraph and what it adds to the paper. Is the paragraph needed? Or could it perhaps be made shorter?

Lines 349-360: These recommendations (especially regarding pensions) seem a little outside the scope of the research and do not directly follow from the main findings on air pollution. Can you more directly link the paragraph to your findings?

Lines 392-393: What is meant by "necessary measures to adjust the population structure"?

Materials and Methods

Lines 439-442. More detail would be helpful on how CMIP6 estimates PM2.5 levels, since this is central to the study.

Lines 478-488: Do the inputs described here all come from CMIP6? Can you comment on the basis for the formula from Chowdhury et al and Van Donkelaar et al?

Reviewer #2 (Remarks to the Author):

The present manuscript presents an interesting study assessing the mortality impacts of PM2.5 in China under different mitigation scenarios and population development trajectories. Differently to previous assessments, authors aimed at exploring the role of several drivers, including ageing, population growth and changes in PM2.5 exposure. They applied established methods to perform the analysis, mostly applied in global burden of disease studies.

I believe that the findings are of great relevance and provide additional insights on the potential benefits of mitigation strategies.

Although the study is well written, given the complexity of the topic (mainly the contribution of drivers), I believe that text and figures could be improved to ease the interpretation of the findings. Thus, overall, my suggestions are aimed to help improving the presentation of the results and clarify some issues related to the contribution of each driver.

Presentation of the results:

- I would suggest the authors to provide, very briefly, additional information on the method and data used before the presentation of the results. In particular, it is difficult to understand the meaning of the different drivers with no prior explanation on the nature or how these are defined in the analysis. For example, how the authors disentangled the contribution of population growth from aging (changes in population structure)? I believe it would be also beneficial to clarify what each driver means.

Results on the drivers:

- I personally see that Figure 3 is not very clear. I wonder if it would be better to just report in bars the contribution of each (over the same x axis) for each time point.

- The explanation of the findings is not clear either: for example, the authors start saying that the main contributor to the increase in DAPP is aging, but in the previous section they mentioned that DAPP decreases or remains similar in all scenarios. In the discussion this statement is even more prominent - " Our study found that population aging will be the primary factor driving an increase in DAPP from 2017-2035". In my opinion, this is not accurate, and probably a more appropriate way to formulate this statement would be that ageing would attenuate the benefits of mitigation

strategies by reducing or counteracting the decrease in DAPP. I understand that the addition of the other components will give a decreasing pattern, but I am sure the authors can find another way to describe the findings.

- Figure 6: although I could find it useful, it is a bit too complex and I would probably try either to simplify or provide more information in the same figure and the legend.

- I would suggest moving Figure 4 into suppl file.

Other general comments:

- Why the authors use specific years to report the findings, instead of time intervals (e.g., 5-year interval)? this approach would smooth out the influence of abrupt changes due to data artifacts.

- Limitations of the study: the authors do not discuss potential limitations of their study. For example, the use of constant association estimates (is it possible that in the future pop would reduce their vulnerability to PM2.5?)

- In Figure 5 authors report the contribution of older pop in DAPP for each cause, but they do not report the actual change in DAPP for each cause (or it was not evident to me).

- Introduction: Paragraph starting in line 66: when the authors describe the existing papers, it would be useful to provide an overview of the findings in previous assessments.

- It is important to highlight that "elderly" is currently not used - better older population or older adults.

- Figure 2: the size of the bars is so small that it is very difficult to see it in the printed version. I would suggest changing the format or removing it (maybe include a table in suppl file).

Reviewer #3 (Remarks to the Author):

The manuscript brings an analysis of the deaths attributable to the PM2.5 population in different provinces in China under future climate projections based on the CMIP6 scenarios.

The study is important, the aging of the population is already and will become even more challenging in the future, concerning health and well-being.

The effect of pollutants, and especially PM, on human health is also a product of many publications and reports. Studies have shown deleterious effects in the elderly.

The different provinces in China constitute an important case study, as they face air pollution, PM levels, high number of habitants, and aging of the population.

But, in my opinion, some aspects have to be considered in the text.

Starting with the abstract I think the language and the results should be presented with more details, for instance, the results presented with a variation of 31.9 and 178.8 in the DAPP could be better related to the scenarios. There are three scenarios presented, from less radiative impact to the highest impact scenario, how they are related to the findings. The numbers don't seem consistent with the results presented in the results section.

In the introduction, the first sentence should be contextualized. The authors wrote that "The air pollution by PM2.5 has increased", but this happens at some locations. The PM2.5 concentrations at different places are only above the new air quality guidelines from WHO.

As the subject of the manuscript is related to the aging of population, it would be adequate to include references on the health effect of PM2.5 on the elderly. There are references of studies around the world discussing the impact of pollution on the elderly.

The sentence "d climate change (e.g., transformation and diffusion of air pollution)" is not clear. Which climate change parameters will affect the diffusion of air pollution. It is too vague.

Before the results more details are needed related to the methodology:

- Which are the driving factors considered? They are presented in the supplementary material but the decomposition of the factors could be discussed in the results.

In line 202 and even before the authors discussed the decreasing trend in the disease mortality (what is really expected), but could this be outshined by the aging of the elderly population?

In my view a discussion about the meteorological conditions in the scenarios and their synergic effects should be presented. The scenarios indicate the increase in the number, duration and intensity of the heat waves, how will these conditions affect the elderly population? This is not pertinente to be discussed?

My last comment is related to the legends of the figures. They need to be improved, not all the information to understand them are presented.

For instance Figure 3 presents the different factors to DAPP changes, but DAPP is one of the variables presented with the other factors. I think it is not clear what is being presented. In the discussion the model KAYA needs a reference. Also the recommendation to activities indoors when there are high levels of pollutants can not work if the indoors ambients are contaminated with high levels of pollutants, as occurs with the use of solid fuels for cooking, for instance.

Reviewer #4 (Remarks to the Author):

This paper makes forecasts of mortality attributable to PM2.5 exposure between 2017 and 2035 and decomposes the change in these deaths by the effects of changes in PM2.5 exposure, population growth, population ageing and mortality trends for causes affected by PM2.5. I'll concentrate on commenting on the areas of my expertise: mortality and population estimation. The conclusion that ageing is the main driver of change in mortality attributable to PM2.5 is hardly surprising as the main diseases affected by PM2.5 have a steep age gradient and China's population is rapidly ageing.

Cause-specific mortality rates were taken from GBD 2017. A question is why not from the more recent version of GBD2019? These mortality rates were forecast using a relatively crude method that was used in the earliest GBD forecasts. Looking at Figure 1 I have a number of questions/comments:

1. age-standardised (as stated in note, not in y-axis title; also no information on what standard was used) rates of IHD is forecast to be flat between 2017 and 2035 while stroke is forecast to continue a fast decline over the same period. I suspect this is an artefact of ignoring the pattern of a rise until 2010 and then a decline. As IHD and stroke share many risk factors, it is unlikely that this diverging pattern will actually occur. More recently in GBD forecasts are driven by forecasts of exposure to risk factors for the risk-dependent part of mortality rates and the older method used in this paper is similar to what is being applied to the 'risk-deleted' proportion of cause-specific mortality rates.
2. I see no reference to forecasts of cause specific mortality being constrained by forecasts of total mortality rates. We tend to have greater faith in forecasts of all-cause mortality because data sources are less prone to measurement bias than those relying on a mix of physician-certified and coded deaths and verbal autopsy methods. This is an important step as separate forecasts cause by cause when aggregated by all causes can lead to large departures from the forecast of all deaths. I suspect the authors have not worried about this by forecasting just 6 causes of death.
3. the panel on % elderly over 65 defies logic. How could the blue and yellow scenarios lead to such large differences? This would need to come from change in fertility, mortality or migration. In/out migration in China is small and unlikely to be a factor. Fertility could affect the denominator but would have to vary by an implausible amount to explain the differences in the panel. Similarly, mortality rates are unlikely to be able to create such a large variation in % 65+. I see similar change (in opposite direction by scenario) in total population. That makes me think there are large differences in assumptions on migration/fertility/mortality underlying these population estimates. However, the paper does not explain the methods of the population forecasts but refers to two publications. Ref 47 is a paper in a journal called Climate Change Research which I was unable to find online. Ref 48 has incomplete information (i.e. no journal listed) but I was able to find it. It seems that very different assumptions on fertility are the main reason for the variation. The high fertility assumptions in SSPs 2 and 3 have not (yet) been borne out after abandoning the one child policy, suggesting a return to much higher fertility is not so likely.
4. the population forecasts are based on assumptions on all-cause mortality that are not consistent with those from which the cause specific mortality rates have been derived (GBD).

Specific comments:

Lines 25-27 your are not convincing me with this statement. Why?

DAPP is an unusual acronym and best avoided

line 29: 'disease mortalities'; an awkward term; did you mean causes of death?

line 27-30: this sentence is a non-sequitur to me.

line 34: in an abstract at first mention of decomposition, I would expect to see the components of

the decomposition

Line 38 (and repeater further on): with net decrease in PM2.5 attributable deaths in most scenarios, the word offset is misleading. What you mean is that ageing partially counters the effects of other factors (particularly lower forecasts of cause-specific mortality rates)

Line 39: that is a big statement to make about the need for improving health care. If you make this argument, I would like to see in much greater detail how you would envisage this being beneficial. Reducing exposure to PM2.5 would need to be the primary policy concern. Is your argument that we need to prevent the six diseases affected by PM2.5 by other means to make it less of a problem from PM2.5? If so, what would you suggest?

Line 47-48: ref 1 makes no mention of PM2.5 at all. Apart from that, if you make such a statement I would like to see more detail: where, by how much and over what period (not just saying 'recent').

first paragraph of introduction: why discuss global issues in a paper that is about China?

Line 68-69: medical conditions as an example of socio-economic development factors? mention of air pollution emissions is also odd as you are basically stating the obvious: 'DAPP' are affected by air pollution

line 77: 'changes in medical conditions' Why mention this? Are you referring to trends in these conditions due to factors other than air pollution? If so, make that clearer.

line 86: Wu may not have used these but GBD produced province level estimates

line 113 (and mentions elsewhere): the terms is comparative risk assessment, not comparable risk assessment

line 121 and following: you do not explain the various SSP scenarios. Many readers will not be familiar with that detail.

Figure 2: presenting counts in units of ten thousands is rather unusual

Line 251: aging the leading challenge? Over the forecast period of this paper, there is little to be done about that. Flagging it as a challenge implies you are able to do something about. I would argue that ability is rather limited.

Line 518: I presume you mean 1000 curve fits and not '1000th curve fit'

line 551: first mention of GBDMAPS without any explanation or reference

Response to the Editor and Reviewers

We would like to express our gratitude to the editor and anonymous reviewers for their valuable comments and suggestions for improving the quality of the paper. We have carefully considered all the points raised by them. We are providing detailed point-by-point responses to all questions and recommendations by the reviewers. In the responses below, red fonts are the revised texts.

Part 1: Response to the reviewer 1

Thank you for giving me the opportunity to review this manuscript. It seems to be a well designed and well performed piece of research on an important topic. The methods appear to be sound and I would recommend publication with relatively minor changes/clarifications.

Issue 1: The title of the paper is slightly confusing. How about “The challenge of population aging for future mitigation of deaths attributable to PM_{2.5} air pollution in China”?

Response: Revised as suggested.

Thank you for your recognition. We have revised the title as suggested.

Issue 2: Minor point: I think it would be better to use “air pollution” rather than just “pollution” throughout (e.g. in the title).

Response: Revised as suggested.

We have changed pollution to air pollution, and revised the title as well as the texts in the manuscript.

Issue 3: Minor point: I’m not keen on creating new acronyms, such as DAPP. As far as I’m aware, this isn’t a common term used in the air pollution field (other than in past papers by the authors).

Response: Revised and clarified.

We tried to avoid using acronym in the revised manuscript, but it strongly affected readability and word limit, especially in the Results and Methods sections. The revised sentences are ridiculously long. Therefore, we are still using the acronym, DAPP, in the revised manuscript.

Issue 4: Abstract. It would be useful to briefly state what the scenarios represent. You might just say e.g. “scenarios representing different levels of sustainable development”.

Response: Revised as suggested.

In the abstract, we briefly described the four scenarios as suggested. In *Materials and methods*, we supplemented more details of these four scenarios. The sentence in the Abstract is revised as below (lines 30-32):

“Here, we estimated spatiotemporal changes in deaths attributable to PM_{2.5} air pollution in China from 2000 to 2035 based on a risk assessment framework and the latest population and climate projections under four scenarios representing different levels of sustainable development (the sustainability (SSP1-2.6), the middle of the road (SSP2-4.5), the regional rivalry (SSP3-7.0) and the fossil-fueled development (SSP5-8.5) scenarios)”

Issue 5: The abstract is a little confusing because it first says “The results show that from 2017 to 2035, DAPP in China will decrease...”, then the next sentence says “We found that population aging will be the leading driver contributing to the increase in DAPP...”. Should the second sentence say something like “We found that population aging will be the leading driver tending to increase DAPP...”?

Response: Revised as suggested. The revised version is shown (lines 35-38) as follows:

“We found that population aging will be the leading contributor to increased deaths attributable to PM_{2.5} air pollution and will counter the positive gains achieved by improvements in air pollution and healthcare”

Issue 6: Introduction. Lines 47-51: Are these lines referring specifically to China? Or global?

Response: Clarified.

These lines are descriptions of the global context. To make the presentation clearer, we have further revised the texts to emphasize that PM_{2.5} air pollution is one of the most important global health threats.

The revised version is shown (lines 45-49) as follows:

“Fine particulate matter (PM_{2.5}) has become one of the most ubiquitous global air pollutants¹⁻³. Exposure to PM_{2.5} air pollution has adverse effects on human health, causing premature death via diseases such as pulmonary and cardiovascular disease^{4,5}. The number of deaths attributable to PM_{2.5} air pollution (DAPP) globally has also increased rapidly, from 1.14 million to nearly 3 million between 2000 and 2017⁶, becoming the fifth-leading mortality risk factor⁷.”

Issue 7: Lines 92-95: It might be worth emphasising here that the RCPs represent greenhouse gas concentration trajectories (as opposed to air pollutants).

Response: Revised as suggested.

We added an additional note on RCPs (lines 92-93), stating that “Specifically, RCPs are multi-model global scenarios of greenhouse gases”

Issue 8: Line 111: Perhaps clearer to say “...impact of population aging on DAPP in China...”.

Response: Revised as suggested.

We rewrote the sentence as below to make it clearer (lines 104-107):

“The purpose of this study is to estimate the temporal and spatial patterns of DAPP based on the CMIP6 dataset, provincial-level disease mortality data and decomposition analysis, and investigate the impact of population aging on them in China under alternative future scenarios considering both socioeconomic development and climate change.”

Issue 9: Results. Lines 161-163: I don't understand the sentence starting “In addition, it is worth noting...”. Please can the authors clarify?

Response: Revised and clarified.

We rewrote this sentence to avoid confusion. The revised version is shown (lines 168-172) as follows (lines 155-158):

“In terms of total change in DAPP, the only increase (0.1%, 0%-0.13%) in DAPP from 2017 to 2035 was projected to occur in Northeast China, while in the six other regions DAPP declined. Central China showed the largest decrease (31%, 22.3% - 38.6%), while the eastern regions showed the smallest (8.8%, 3.2% - 10.7%, 95% CI).”

Issue 10: Lines 182-185: I also find the sentence here confusing: “In the sustainability scenario (SSP1-2.6), ...”.

Response: Revised and clarified.

We rewrote this sentence to avoid confusion. The revised version is shown (lines 159-162) as follows:

“In the sustainability scenario (SSP1-2.6), the decrease in DAPP in both province is more than 4 times to the national average (i.e., the average of 31 provinces) in China, and under the regional rivalry scenario (SSP3-7.0) scenario is more than 14 times the national average.”

Issue 11: Lines 187-188: Like the abstract, this sentence is a little confusing because it suggests DAPP increases from 2017 to 2035.

Response: Revised and clarified.

We rewrote the sentence to avoid confusion. The revised version is shown as follows (lines 174-178):

“Using the decomposition method (see Materials and methods for details), we compared the effects of the four factors that contribute to the change of DAPP (Figure 3). Among the four factors, population aging was the only factor contributing to the growth of DAPP in the sustainability (SSP1-2.6) and fossil-fueled development (SSP5-8.5) scenarios, resulting in increases of 674 thousand and 703 thousand deaths per annum, respectively.”

Issue 12: Figure 3: It would be helpful to the reader to explain briefly how to read/interpret the plots (either in the figure title or in the main text).

Response: Revised as suggested.

We have revised the Figure 3 added the instructions in Figure 3 to help readers better understand the role of factors such as age structure in affecting the DAPP and its relationship with changes in total deaths. The revised version is shown (lines 186-189) as follows:

Figure 3 Contributions of different factors to deaths attributable to PM_{2.5} air pollution (DAPP) changes in China from 2017 to 2035. Note, these plots show the cumulative effect of four factors: age structure, total population, air quality, and disease mortality, on the DAPP between 2017-2025 and 2025-2035.

Meanwhile, considering that the previous description may confuse readers, we have also added and updated the description of this figure in the texts as follows (lines 174-178):

“Using the decomposition method ^{7,10} (see *Materials and methods* for details), we compared the effects of the four factors that contribute to the change of DAPP (Figure 3). Among the four factors, population aging was the only factor contributing to the growth of DAPP in the *sustainability* (SSP1-2.6) and *fossil-fueled development* (SSP5-8.5) scenarios, resulting in increases of 674 thousand and 703 thousand deaths per annum, respectively..”

Issue 13: Discussion. Line 259: I don't think the paper by Wang et al was mentioned in the

Introduction when making the case for the research. Can the authors explain how their study compares (and improves upon) the Wang et al study?

Response: Revised and clarified.

Firstly, we have added the research of Wang et al. in the Introduction as follows (lines 82-88):

“Wang et al. estimated the health burden of DAPP in 2020 and 2030 based on air quality scenarios and population trends under the Shared Socioeconomic Pathways (SSP)¹¹. Although the study estimated the impact of population aging and other factors on DAPP in China based on future socioeconomic changes, it did not consider the changes in PM_{2.5} concentration in the context of climate change.”

Compared with the study of Wang et al., we have two main advantages. First of all, from the perspective of scenario setting, Wang et al. mainly considered future socioeconomic changes and did not consider the future climate change factors when estimating future DAPP in China. Our study addressed this research gap by comprehensively considering the impact of future climate change and socioeconomic change on DAPP based on the SSPs-RCP scenario combination. Second, from the time scale of the study, Wang et al. estimated DAPP in China in 2020 and 2030. Our paper estimates China's deaths DAPP from 2017 to 2035. Therefore, we believe our study improve upon Wang et al..

Issue 14: Lines 272-286: The start of the paragraph suggests that “two aspects” will be discussed but the paragraph seems to make three distinct points (“Third, with the fast-growing aging population in China, ...”).

Response: Revised as suggested.

We have modified it and added one more point. The revised version is shown as follows (lines 242-243):

“the DAPP-related challenges posed by population aging are mainly demonstrated in the following four aspects.”

Issue 15: Lines 289-292: The KAYA model needs a little further brief description and a reference. What does it do?

Response: Revised and clarified

We have deleted the quantitative results on the mediating effects of population ageing on DAPP based on the KAYA model here for several reasons. First, DAPP were estimated based upon population and population structure, which are not appropriate for treating as the dependent variable in the KAYA model. Second, when we look into the literature and found that previous studies have not reached a consensus on the mediating effect of population ageing on the deaths. Some results supported this statement (Estiri and Zagheni, 2019; Loi and Loo, 2016; Zha et al., 2022) while others opposed this claim (Yang et al., 2018). Therefore, we deleted the KAYA model results and remained a qualitative discussion on the association between populating ageing and DAPP based on previous literature as follows (line 264-271).

“In addition to directly causing the DAPP, the aging of the population may also affect DAPP through their energy consumption behavior and household consumption structure, and corresponding air pollutant emissions. Existing knowledge on the associations between population ageing and air pollutants are mixed and piecemeal. For example, some studies found that older people may consume more fossil energy^{21,36} and energy intensive products and services³⁷, while others found that older people may consume less fossil energy³⁸. In the context of an aging society³⁹, how different age groups’ energy consumption behaviors and household consumption structure affect air pollution needs further investigation.”

Added references:

Estiri, Hossein, and Emilio Zagheni. "Age matters: Ageing and household energy demand in the United States." *Energy Research & Social Science* 55 (2019): 62-70.

Loi TS, Loo SL (2016) The impact of Singapore’s residential electricity conservation efforts and the way forward. Insights from the bounds testing approach. *Energy Policy* 98:735–743. <https://doi.org/10.1016/j.enpol.2016.02.045>

Yang Y, Deng YR, Chen FF (2018) Impact of population aging and industrial structure on CO2 Emissions and emissions trend prediction in China. *Atmos Pollut Res* 9:446–454. <https://doi.org/10.1016/j.apr.2017.11.008>

Zha, D., Liu, P. & Shi, H. Does population aging aggravate air pollution in China? Mitigation and Adaptation Strategies for Global Change 27, doi:10.1007/s11027-021-09993-y (2022).

Issue 16: Lines 318-339: I’m not entirely sure what point is being made in this paragraph and what it adds to the paper. Is the paragraph needed? Or could it perhaps be made shorter?

Response: Revised as suggested.

We have made this paragraph shorter to be briefer and more concise (lines 345-350). The revised version is shown as follows (lines 289-293):

“The difficulty in preventing and mitigating DAPP has regional differences in China. Our results showed that, while Eastern, Southeast and Southwestern China have made some positive progresses in preventing air pollution and improving health care^{12,13}, there is still room to improve air quality and health care in Central, West and Northeast China with a rapidly aging society¹⁴. Such regional differences also echo with the challenges faced by countries with varying developing levels (Table 1)”

Issue 17: Lines 349-360: These recommendations (especially regarding pensions) seem a little outside the scope of the research and do not directly follow from the main findings on air pollution. Can you more directly link the paragraph to your findings?

Response: Revised as suggested.

We have simplified and rewritten this section. We removed the discussion on pension and more directly link the discussion to our findings. The revised texts are as follows (lines 301-307):

“Second, for regions that have achieved greater economic development (e.g., Northeast China), the threat of population aging was the main driver of increased DAPP. For these regions, providing healthcare and improving basic pension plans for older people is a priority, for instance, establishing long-term nursing systems to meet the requirements of older people⁴⁵. We also note that the air pollutants generated in Northeast China may lead to haze episodes in neighboring provinces⁴⁶ (e.g., Beijing-Tianjin-Hebei urban agglomeration) via atmospheric pollution transport. These spatial spillover effects require collaboration among regions to mitigate the adverse effect of air pollution.”

Issue 18: Lines 392-393: What is meant by “necessary measures to adjust the population structure”?

Response: Revised and clarified.

We rewrote these sentences to avoid confusion. Although fertility can be adjusted through policies (such as incentives to have children, and others), changing the age structure of the population is indeed an unattainable goal considering the short study period of this paper (i.e., 2017-2035) and global trend. Therefore, we believe that it is more important to proactively deal with population aging. This includes providing targeted medical services for the elderly population and building an age-friendly society. Based on this, we have removed this sentence and replaced with other texts as follows (lines 320-323):

“To improve the health and well-being of residents, relevant policies should consider the aging population. At the government level, the consideration of aging needs to be fully integrated into health policies to build an age-friendly society. In addition, it is imperative to increase the investment in healthcare to reduce the economic burden of older people”

Issue 19: Materials and Methods. Lines 439-442. More detail would be helpful on how CMIP6 estimates PM2.5 levels, since this is central to the study.

Response: Revised as suggested.

We have added more details of scenario settings in the *Materials and Methods* section, and described the different scenarios of CMIP6 in more detail. The revised version is shown as follows (lines 371-389):

“ScenarioMIP classifies the scenarios of future socioeconomic and climate change into eight primary and secondary scenarios based on their priorities^{19,58}. In this study, we selected four scenarios with high priority in the first-level experiment to estimate DAPP. Among them, SSP1-2.6 is a sustainable green path characterized by low vulnerability and low mitigation challenges, with a mild climatic change and a small range of variation^{19,26}. In this scenario, the birth rate, death rate and migration rate are lower; population is more educated; and population is older¹⁶. SSP2-4.5 is an intermediate path with a moderate climatic change characterized by moderate social vulnerability

and a moderate climatic change. In this scenario, social, economic and technological trends are similar to historical ones¹⁶, so that the changes in fertility, mortality, migration and the proportion of the older population also maintain current trends. SSP3-7.0 is a combined scenario of regional competing paths characterized by higher social vulnerability and higher mitigation challenges with a moderate to severe climatic change. In this scenario, population grows rapidly; the regional development is unbalanced, and the per capita economic and technological development level is low¹⁶. With rapid growth of population size, the fertility rate and mortality rate are relatively high, and the degree of population aging is relatively low²⁶. SSP5-8.5 is a scenario dominated by traditional fossil fuel combustion, characterized by high social vulnerability and high mitigation challenges, with a severe climatic change. In this scenario, social development will emphasize economic growth, while the fertility rate, mortality rate and migration rate are relatively low; and the degree of population aging is relatively high²⁶.”

Issue 20: Lines 478-488: Do the inputs described here all come from CMIP6? Can you comment on the basis for the formula from Chowdhury et al and Van Donkelaar et al?

Response: Clarified.

Yes. Future pollutant concentration data are from CMIP6 (<https://pcmdi.llnl.gov/CMIP6/>), which mainly contains multiple climate model simulations of sulfate, black carbon, organic aerosols, dust and sea salt in the surface, with a spatial resolution of approximately 1 degree, The temporal resolution is monthly from 2015 to 2100. These concentrations of pollutants are driven by the emission, meteorological field and underlying surface corresponding to each scenario in ScenarioMIP, and the results are logically and inherently consistent with the socioeconomic factors underlying each SSPs²¹.

Chowdhury et al and Van Donkelaar et al estimated future concentrations of PM_{2.5} based on empirical formula, but the estimations are often lower than those derived from remote sensing data and monitoring data²². In addition, the estimated concentrations obtained through the empirical formula mainly reflect the long-term trend of PM_{2.5} concentration, and its interannual variation is more smoother than observed variations^{22,23}. Therefore, we further corrected the estimated results based on historical monitoring data of PM_{2.5} concentration. The specific formula is as follows²¹:

$$PM_{2.5f}^{cal} = PM_{2.5bl}^{obs} \times \frac{PM_{2.5f}^{est}}{PM_{2.5bl}^{est}}$$

where $PM_{2.5f}^{cal}$ represents the calibrated PM_{2.5} concentration in the future; $PM_{2.5bl}^{obs}$ represents the observed PM_{2.5} concentration during the baseline period (2012-2017), and $PM_{2.5bl}^{est}$ represents corresponding estimated PM_{2.5} concentration during the baseline period obtained from the empirical equation. $PM_{2.5f}^{est}$ represents the estimated PM_{2.5} concentration in the future using the empirical equation.

Part 2: Response to the reviewer 2

Issue 1: The present manuscript presents an interesting study assessing the mortality impacts of PM2.5 in China under different mitigation scenarios and population development trajectories. Differently to previous assessments, authors aimed at exploring the role of several drivers, including ageing, population growth and changes in PM2.5 exposure. They applied established methods to perform the analysis, mostly applied in global burden of disease studies.

I believe that the findings are of great relevance and provide additional insights on the potential benefits of mitigation strategies.

Although the study is well written, given the complexity of the topic (mainly the contribution of drivers), I believe that text and figures could be improved to ease the interpretation of the findings. Thus, overall, my suggestions are aimed to help improving the presentation of the results and clarify some issues related to the contribution of each driver.

Response: Thank you for the recognition. We have revised the manuscript according to your comments and suggestions. Please check our responses as below.

Issue 2: Presentation of the results:

- I would suggest the authors to provide, very briefly, additional information on the method and data used before the presentation of the results. In particular, it is difficult to understand the meaning of the different drivers with no prior explanation on the nature or how these are defined in the analysis. For example, how the authors disentangled the contribution of population growth from aging (changes in population structure)? I believe it would be also beneficial to clarify what each driver means.

Response: Revised as suggested.

Before describing the deconstruction results, we have briefly summarized the main research methods and explained how different drivers affect DAPP, which can help readers to better understand this section. The revised version is shown as follows (lines 174-178):

“Using the decomposition method ^{7,10} (see *Materials and methods* for details), we compared the effects of the four factors that contribute to the change of DAPP (Figure 3). Among the four factors, population aging is the only factor which may contribute to the growth of DAPP in the *sustainability* (SSP1-2.6) and *fossil-fueled development* (SSP5-8.5) scenarios, resulting in increases of 674 thousand and 703 thousand deaths per annum, respectively.”

Issue 3: Results on the drivers:

I personally see that Figure 3 is not very clear. I wonder if it would be better to just report in bars the contribution of each (over the same x axis) for each time point.

Response: Revised as suggested.

We revised this figure to make it clearer. In the revised figure, we only describe the contribution of four drivers to deaths attributable to PM_{2.5} air pollution over a period of time (i.e., 2017-2025 and 2025-2035). We modified the x axis of the graph to clarify the two time periods. We have also added the overall changes of the four drivers at each stage more clearly in the figure. We also added the illustration of the diagram to help readers digest the diagram. The revised version is shown (lines 195-200) as follows:

Figure 3 Contributions of different factors to **deaths attributable to PM_{2.5} air pollution (DAPP)** changes in China from 2017 to 2035. Note, these plots show the cumulative effect of four factors: age structure, total population, air quality, and disease mortality, on the **DAPP** between 2017-2025 and 2025-2035.

Issue 4: The explanation of the findings is not clear either: for example, the authors start saying that the main contributor to the increase in DAPP is aging, but in the previous section they mentioned that DAPP decreases or remains similar in all scenarios. In the discussion this statement is even more prominent - " Our study found that population aging will be the primary factor driving an increase in DAPP from 2017-2035". In my opinion, this is not accurate, and probably a more appropriate way to formulate this statement would be that ageing would attenuate the benefits of mitigation strategies by reducing or counteracting the decrease in DAPP. I

understand that the addition of the other components will give a decreasing pattern, but I am sure the authors can find another way to describe the findings.

Response: Revised as suggested.

Sorry for the confusing descriptions. We have revised the descriptions of the results. The revised texts are as follows (lines 190-196):

“From 2025 to 2035, the age structure was the most important factor driving the increase in DAPP. The increase in the older population has led to an increase of 373 (342-398) thousand DAPP, an increase of 69(57-91) thousand from 2017 to 2025. The impact of disease mortality on DAPP was projected to level off. From 2025 to 2035, the reduction in disease mortality reduced DAPP by 266 (238-276) thousand DAPP, which is 83 (29-56) thousand less than 2017-2025. The impact of changing age structure on DAPP began to exceed the impact of declining disease mortality on DAPP and became the main reason for the increase in DAPP from 2025-2035.”

Meanwhile, in the Abstract and other parts of the article, we also updated the relevant description of this part. The specific contents are as follows (lines 36-37):

“We found that population aging will be the leading driver contributing to increased deaths attributable to PM_{2.5} air pollution and will counter the positive gains achieved by improvements in air pollution and healthcare.”

Issue 5: Figure 6: although I could find it useful, it is a bit too complex and I would probably try either to simplify or provide more information in the same figure and the legend.

Response: Revised as suggested.

We have revised this figure and its caption to help the readers understand this figure (lines 253-255):

Figure 5 Contribution of age structure change to deaths attributable to PM_{2.5} air pollution among the seven regions of China across four scenarios.

Issue 6: I would suggest moving Figure 4 into suppl file.

Response: Revised as suggested.

We have moved Figure 4 into the supplementary files as supplementary Figure S4.

Issue 7: Other general comments: Why the authors use specific years to report the findings, instead of time intervals (e.g., 5-year interval)? this approach would smooth out the influence of abrupt changes due to data artifacts.

Response: Revised and clarified

In fact, we used the five-year intervals to describe the results when we plot the figure (lines 129).

We also added the explanation in the Methods and Results section (lines 508-509). The revised version is shown as follows:

“In addition, we reported the 5-year intervals to smooth out the influence of abrupt changes due to data artifacts (Figure 1 and Figure 2).”

Figure 1. Projected change in driving factors from 2000-2035. (d-i) Age-standardized disease mortality. Note that the age structure of the sixth census in China was used as the standard age structure²⁸. We used 5-year smoothed time intervals to visualize the results.

Issue 8: Limitations of the study: the authors do not discuss potential limitations of their study. For example, the use of constant association estimates (is it possible that in the future pop would reduce their vulnerability to PM2.5?)

Response: Revised as suggested.

We have added Section 3.3 to discuss the limitations of the study caused by model selection, future population vulnerability changes and other factors (lines 393-409). The revised version is shown as follows (lines 325-340):

“There are several sources of uncertainty and limitations in our research. First, we adopted the MR-BRT model developed by the World Health Organization (WHO) and Global Burden of Disease (GBD). To assess the impact of this choice we compared the results of this study with deaths calculated by Geng et al²⁵ based on the Global Exposure Mortality Model (GEMM), which models all-natural cause mortality and does not incorporate additional types of exposure, such as active smoking²⁶. Although the absolute values of the two results differ, the trends are consistent.

Second, we assumed a constant association estimate for population vulnerability to PM_{2.5} air pollution. The results of this comparison suggest that the combined effects of air pollution and other risk factors (such as extreme temperature) on some diseases and the change in vulnerability to air pollution in our study may be conservative²⁷⁻²⁹. Third, when estimating the health burden of air pollution, we considered both the effects of climate change and socioeconomic factors, but there are still some other factors that remain difficult to take into account. For example, we did not consider future policy interventions or indoor air pollution. Future public emergencies (e.g., pandemic) or possible policies (e.g., emissions reduction measures and temporary lockdowns) will affect the exposure of air pollution. Therefore, the projections in this study only reflect long-term trends in the health effects of outdoor PM_{2.5} air pollution.”

Issue 9: In Figure 5 authors report the contribution of older pop in DAPP for each cause, but they do not report the actual change in DAPP for each cause (or it was not evident to me).

Response: Revised as suggested.

We supplemented the Figure (supplementary Figure S5) to report the actual change in deaths attributable to PM_{2.5} air pollution for each cause.

Figure S5 Deaths attributable to PM_{2.5} air pollution by disease from 2000 to 2035. Note that COPD, DM2, IHD, LC, and LRI refer to chronic obstructive pulmonary disease, type 2 diabetes, ischemic heart disease, lung cancer, and lower respiratory tract infection, respectively. Shading indicates the 95% confidence interval.

Issue 10: Introduction: Paragraph starting in line 66: when the authors describe the existing papers, it would be useful to provide an overview of the findings in previous assessments.

Response: Revised as suggested.

We have provided an overview of the findings in previous assessments. The revised version is shown as follows (lines 71-73):

“Several studies have analyzed the relationship between PM_{2.5} air pollution and the older adult population in China, mainly based on historical data to analyze the impact of population aging on a specific disease³⁰⁻³² (e.g. respiratory disease or heart disease).”

Issue 11: It is important to highlight that "elderly" is currently not used - better older population or older adults.

Response: Revised as suggested.

We have changed the term throughout the manuscript.

Issue 12: Figure 2: the size of the bars is so small that it is very difficult to see it in the printed version. I would suggest changing the format or removing it (maybe include a table in suppl file).

Response: Revised as suggested.

We have modified the size of the picture, and added the table (TableS2) in the supplementary materials. The revised version is shown as follows (lines 169-172):

Figure 2 Varying trends in deaths attributable to PM_{2.5} air pollution (DAPP) from 2000 to 2035. Note: changes in the total amount of DAPP (a) in China and among regions (c), and changes in DAPP per capita (b) in China and among regions (d). The values for each region are the average estimates among the provinces in this region (see Table S2).

Table S2 Deaths attributable to PM_{2.5} air pollution among regions in China(a) Deaths attributable to PM_{2.5} air pollution (thousand people)

Area Year	North	Northeast	East	Central	South	Southwest	Northwest
SSP1-2.6							
2000	17.25	17.9	31.4	52.97	22.34	22.03	13.82
2005	19.52	22.26	34.59	56.21	28.59	26.53	14.83
2010	19.52	22.26	34.59	56.21	28.59	26.53	14.83
2015	23.89	25.5	35.55	62.92	29.12	23.61	15.48
2020	20.7	28.2	35.58	54.85	26.79	23.4	12.54
2025	19.56	26.69	32.7	49.56	25.02	20.76	12.09
2030	18.83	26	32.46	46.62	24.58	19.42	12.18
2035	18.49	25.31	32.08	43.9	24.66	17.56	11.88
SSP2-4.5							
2000	17.25	17.9	31.4	52.97	22.34	22.03	13.82
2005	19.52	22.26	34.59	56.21	28.59	26.53	14.83
2010	19.52	22.26	34.59	56.21	28.59	26.53	14.83
2015	23.89	25.5	35.55	62.92	29.12	23.61	15.48
2020	20.99	28.92	36.47	56.12	27.58	24.28	12.64
2025	21.06	29.07	35.43	53.44	26.66	22.32	12.66
2030	21.62	29.79	37.13	53.49	28.2	22.59	13.29
2035	22.19	30.36	38.68	52.75	29.38	21.32	13.39
SSP3-7.0							
2000	17.25	17.9	31.4	52.97	22.34	22.03	13.82
2005	19.52	22.26	34.59	56.21	28.59	26.53	14.83
2010	19.52	22.26	34.59	56.21	28.59	26.53	14.83
2015	23.89	25.5	35.55	62.92	29.12	23.61	15.48
2020	21.5	29.33	39.01	56.83	28.31	24.69	12.81
2025	21.86	30.67	38.08	55.55	27.84	23.39	12.76
2030	22.65	32.1	40.84	57.35	29.89	24.12	13.4
2035	23.31	33.45	42.16	58.16	31.23	23.79	13.69
SSP5-8.5							
2000	17.25	17.9	31.4	52.97	22.34	22.03	13.82
2005	19.52	22.26	34.59	56.21	28.59	26.53	14.83
2010	19.52	22.26	34.59	56.21	28.59	26.53	14.83
2015	23.89	25.5	35.55	62.92	29.12	23.61	15.48
2020	20.25	27.68	36.41	54.59	26.4	23.58	12.63
2025	19.32	27.12	33.93	49.89	24.94	21.1	12.05

2030	19.11	26.93	34.78	48.13	26.36	20.46	12.48
2035	19.76	27.32	35.65	46.04	27.08	19.06	12.53

(b) Deaths attributable to PM_{2.5} air pollution (per thousand people)

Area Year	North	Northeast	East	Central	South	Southwest	Northwest
SSP1-2.6							
2000	69.74	68.83	67.63	87.22	58.82	72.07	94.09
2005	69.74	68.83	67.63	87.22	58.82	72.07	94.09
2010	69.41	64.38	60.44	89.4	53.12	70.16	95.4
2015	71.35	76.59	63.1	93.85	54.49	65.65	87.44
2020	66.75	86.69	62.68	82.45	52.01	62.06	67.95
2025	64.11	84.17	57.59	74.81	46.91	55.44	64.51
2030	62.82	84.44	57.67	71.21	44.74	52.03	65.02
2035	63.39	85.6	57.7	67.92	43.68	47.93	63.41
SSP2-4.5							
2000	69.74	68.83	67.63	87.22	58.82	72.07	94.09
2005	69.74	68.83	67.63	87.22	58.82	72.07	94.09
2010	69.41	64.38	60.44	89.4	53.12	70.16	95.4
2015	71.35	76.59	63.1	93.85	54.49	65.65	87.44
2020	67.03	88.16	63.63	83.44	52.85	63.53	67.71
2025	67.82	90.47	61.27	78.97	48.45	58.34	66.23
2030	70.65	95.2	64.54	79.54	49.13	59.04	68.73
2035	74.09	100.62	67.91	78.9	49.13	56.26	68.69
SSP3-7.0							
2000	69.74	68.83	67.63	87.22	58.82	72.07	94.09
2005	69.74	68.83	67.63	87.22	58.82	72.07	94.09
2010	69.41	64.38	60.44	89.4	53.12	70.16	95.4
2015	71.35	76.59	63.1	93.85	54.49	65.65	87.44
2020	69.47	88.72	67.75	83.39	53.04	63.7	67.98
2025	70.73	94.2	65.22	79.98	49.88	59.24	65.54
2030	74.51	101.15	70.34	81.83	51.43	59.88	67.35
2035	78.52	108.77	73.28	81.8	51.42	58.23	67.38
SSP5-8.5							
2000	69.74	68.83	67.63	87.22	58.82	72.07	94.09
2005	69.74	68.83	67.63	87.22	58.82	72.07	94.09
2010	69.41	64.38	60.44	89.4	53.12	70.16	95.4
2015	71.35	76.59	63.1	93.85	54.49	65.65	87.44
2020	64.87	85.21	64.09	82.42	51.28	62.66	68.43
2025	62.17	85.79	59.62	76.2	46.84	56.86	64.56
2030	62.25	87.94	61.53	75.41	47.47	56.09	66.61

Part 3: Response to the reviewer 3

Issue 1: The manuscript brings an analysis of the deaths attributable to the PM2.5 population in different provinces in China under future climate projections based on the CMIP6 scenarios.

The study is important, the aging of the population is already and will become even more challenging in the future, concerning health and well-being.

The effect of pollutants, and especially PM, on human health is also a product of many publications and reports. Studies have shown deleterious effects in the elderly.

The different provinces in China constitute an important case study, as they face air pollution, PM levels, high number of habitants, and aging of the population.

But, in my opinion, some aspects have to be considered in the text.

Response: Thank you for the recognition. We have revised the manuscript according to your comments and suggestions. Please check our responses as below.

Issue 2: Starting with the abstract I think the language and the results should be presented with more details, for instance, the results presented with a variation of 31.9 and 178.8 in the DAPP could be better related to the scenarios. There are three scenarios presented, from less radiative impact to the highest impact scenario, how they are related to the findings. The numbers don't seem consistent with the results presented in the results section.

Response: Revised as suggested.

We have added more details to the summary and corrected the inconsistency. The revised version is shown as follows (lines 33-36):

“The results show that from 2017 to 2035, the deaths in China will decrease between 1.9 (1.6-2.1, 95% CI) thousand under the SSP5-8.5 scenario and 95.1 (84.4-101.6) thousand under the SSP1-2.6 scenario.”

Issue 3: In the introduction, the first sentence should be contextualized. The authors wrote that “The air pollution by PM2.5 has increased”, but this happens at some locations. The PM2.5 concentrations at different places are only above the new air quality guidelines from WHO.

Response: Revised.

We have revised this sentence to avoid the confusion. The revised texts are as follows (lines 45-47):

“Fine particulate matter (PM_{2.5}) has become one of the most ubiquitous global air pollutants¹⁻³. Exposure to PM_{2.5} air pollution has adverse effects on human health, causing premature death via diseases such as pulmonary and cardiovascular disease^{4,5}.”

Issue 4: As the subject of the manuscript is related to the aging of population, it would be adequate to include references on the health effect of PM_{2.5} on the elderly. There are references of studies around the world discussing the impact of pollution on the elderly.

Response: Revised as suggested.

In the Introduction, we added several articles related to population aging and its health burden. In the third paragraph of the Introduction, we focus on the introduction of literature related to population aging. The revised version is shown as follows (lines 70-88):

“Several studies have analyzed the relationship between PM_{2.5} air pollution and the older adult population in China, mainly based on historical data to analyze the impact of population aging on a specific disease²⁰⁻²² (e.g., respiratory disease or heart disease). For example, Chi et al. analyzed the impact of air pollution on the older patients with chronic obstructive pulmonary disease and found that increased PM_{2.5} air pollution during the heating season was associated with a reduction in forced expiratory volume²². Several studies have analyzed historical relationships between population aging and DAPP. For example, Liu et al. estimated the number of deaths caused by air pollution and concluded that population aging caused an additional 950,000 deaths from 2004 to 2017, delaying the reduction of the health burden of PM_{2.5} air pollution²³. Although a few studies have estimated future changes of DAPP, with estimates based upon certain climatic or socioeconomic assumptions. For example, Yue et al. estimated the change of China's DAPP in 2030 under different PM_{2.5} concentration scenarios, but ignored changes in demographics and disease mortality²⁴. Wang et al. estimated the health burden of DAPP in 2020 and 2030 based on air quality scenarios and population trends under the Shared Socioeconomic Pathways (SSPs)²⁵. Although the study estimated the impact of population aging and other factors on DAPP in China based on future socioeconomic changes, it did not consider the changes in PM_{2.5} concentration in the context of climate change. Hence, there is an urgent need for a comprehensive and detailed quantification and projection of DAPP trends in China which ties these complex components together.”

Add references:

Yue, H., He, C., Huang, Q., Yin, D. & Bryan, B. A. Stronger policy required to substantially reduce deaths from PM_{2.5} pollution in China. Nature Communications 11, doi:10.1038/s41467-020-15319-4 (2020).

Murray, C. J. L. & Lopez, A. D. On the comparable quantification of health risks: Lessons from the Global Burden of Disease Study. Epidemiology 10, 594-605, doi:10.1097/00001648-199909000-00029 (1999).

Chi, R. et al. Different health effects of indoor- and outdoor-originated PM_{2.5} on cardiopulmonary function in COPD patients and healthy elderly adults. *Indoor Air* **29**, 192-201, doi:10.1111/ina.12521 (2019).

Liu, Y. et al. Population aging might have delayed the alleviation of China's PM_{2.5} health burden. *Atmospheric Environment* **270**, doi:10.1016/j.atmosenv.2021.118895 (2022).

Wang, Q., Wang, J., Zhou, J., Ban, J. & Li, T. Estimation of PM_{2.5}-associated disease burden in China in 2020 and 2030 using population and air quality scenarios: a modelling study. *Lancet Planetary Health* **3**, E71-E80, doi:10.1016/s2542-5196(18)30277-8 (2019).

Issue 5: The sentence “d climate change (e.g., transformation and diffusion of air pollution)” is not clear. Which climate change parameters will affect the diffusion of air pollution. It is too vague.

Response: Revised as suggested.

We have added the explanation on how climate change affects the spread of air pollution as follows (lines 63-67):

“The number of DAPP is determined by four factors: age structure, total population, PM_{2.5} concentration, and disease mortality⁷. Changes in these four factors are influenced by both climate change and socioeconomic change. For example, the changes in climate conditions such as temperature, humidity, and air pressure can directly affect the secondary transformation and diffusion of PM_{2.5}^{16,17}, thereby changing the atmospheric concentration of PM_{2.5}.”

Issue 6: Before the results more details are needed related to the methodology:

- Which are the driving factors considered? They are presented in the supplementary material but the decomposition of the factors could be discussed in the results.

Response: Revised as suggested.

Before describing the deconstruction results, we have briefly summarized the main research methods and explained how different drivers affect DAPP which can help readers have a better understanding of this part. The revised version is shown as follows (lines 174-178):

“Using the decomposition method^{7,10} (see *Materials and methods* for details), we compared the effects of the four factors that contribute to the change of DAPP (Figure 3). Among the four factors, population aging is the only factor which may contribute to the growth of DAPP in the *sustainability* (SSP1-2.6) and *fossil-fueled development* (SSP5-8.5) scenarios, resulting in increases of 674 thousand and 703 thousand deaths per annum, respectively. In the *middle of the road* (SSP2-4.5) and *regional rivalry* (SSP3-7.0) scenarios, although population growth will also bring a small amount of growth in DAPP, more than 98% of it is due to population aging.”

Also, we have added some decomposition of the factors in the *Introduction* as follows (lines 63-70):

“The number of DAPP is determined by four factors: age structure, total population, PM_{2.5} concentration, and disease mortality⁷. Changes in these four factors are influenced by both climate

change and socioeconomic change. For example, the changes in climate conditions such as temperature, humidity, and air pressure can directly affect the secondary transformation and diffusion of PM_{2.5}^{16,17}, thereby changing the atmospheric concentration of PM_{2.5}. Socio-economic development affects the emission of pollutants and the concentration of PM_{2.5} concentration, as well as the demographic characteristics and the level of available healthcare, which affect the proportion of older people in the population and the mortality rate of diseases^{18,19}.”

Issue 7: In line 202 and even before the authors discussed the decreasing trend in the disease mortality (what is really expected), but could this be outshined by the aging of the elderly population?

Our results showed that reductions in disease mortality will not be offset by increases in population ageing until 2025. The increase in DAPP caused by the aging of the population will be offset by the decrease in disease mortality. Therefore, in general, DAPP in China will show a downward trend from 2017 to 2025. To make our expression more clear, we have made some modifications as follows (lines 190-196):

“From 2025 to 2035, the age structure was the most important factor driving the increase in DAPP. The increase in the older population has led to an increase of 373 (342-398) thousand DAPP, an increase of 69(57-91) thousand from 2017 to 2025. The impact of disease mortality on DAPP was projected to level off. From 2025 to 2035, the reduction in disease mortality reduced DAPP by 266 (238-276) thousand DAPP, which is 83 (29-56) thousand less than 2017-2025. The impact of changing age structure on DAPP began to exceed the impact of declining disease mortality on DAPP and became the main reason for the increase in DAPP from 2025-2035.”

Issue 8: In my view a discussion about the meteorological conditions in the scenarios and their synergic effects should be presented. The scenarios indicate the increase in the number, duration and intensity of the heat waves, how will these conditions affect the elderly population? This is not pertinente to be discussed?

Response: Revised as suggested.

In the Discussion section (*3.1 Population aging is the most important challenge*), we have added the discussion of relevant content and deleted the original content. The revised version is shown as follows (lines 256-263):

“global climate change can also increase the vulnerability of older people in the future. The increased frequency and intensity of extreme events such as heat waves, floods, droughts and wildfires under climate change (especially under the SSP3-7.0 scenario with high population growth and high emissions)³⁶ may also increase the morbidity among older people³⁷. In increasingly extreme climates, older people are more physically and mentally vulnerable and at greater risk of premature death. For example, Yuan et al. showed that, for every 1°C increase in temperature, the health score of older workers (aged 60 and above) decreases by about 6.4 times as much as that of young and middle-aged workers (aged 15-59)³⁸.”

Issue 9: My last comment is related to the legends of the figures. They need to be improved, not all the information to understand them are presented.

For instance Figure 3 presents the different factors to DAPP changes, but DAPP is one of the variables presented with the other factors. I think it is not clear what is being presented.

Response: Revised as suggested.

We revised and improved Figure 3. First, we modified the x axis, which represents an interval of time. In addition, we defined the meaning of each bar. The gray bars represents the value of the DAPP at a given year, while the colored bars represent how the deaths changed over that interval. The revised version is shown as follows (lines 186-189):

Figure 3 Contributions of different factors to deaths attributable to PM_{2.5} air pollution (DAPP) changes in China from 2017 to 2035. Note, these plots show the cumulative effect of four factors: age structure, total population, air quality, and disease mortality, on the DAPP between 2017-2025 and 2025-2035.

Issue 10: In the discussion the model KAYA needs a reference.

Response: Revised and Clarified.

We have deleted the quantitative results on the mediating effects of population ageing on DAPP based on the KAYA model here for several reasons. First, DAPP were estimated based upon

population and population structure, which are not appropriate for treating as the dependent variable in the KAYA model. Second, when we look into the literature and found that previous studies have not reached a consensus on the mediating effect of population ageing on the deaths. Some results supported this statement (Estiri and Zagheni, 2019; Loi and Loo, 2016; Zha et al., 2022) while others opposed this claim (Yang et al., 2018). Therefore, we deleted the KAYA model results and remained a qualitative discussion on the association between populating ageing and DAPP based on previous literature as follows (line 264-271).

“In addition to directly causing the DAPP, the aging of the population may also affect DAPP through their energy consumption behavior and household consumption structure, and corresponding air pollutant emissions. Existing knowledge on the associations between population ageing and air pollutants are mixed and piecemeal. For example, some studies found that older people may consume more fossil energy^{21,36} and energy intensive products and services³⁷, while others found that older people may consume less fossil energy³⁸. In the context of an aging society³⁹, how different age groups’ energy consumption behaviors and household consumption structure affect air pollution needs further investigation.”

Added references:

Estiri, Hossein, and Emilio Zagheni. "Age matters: Ageing and household energy demand in the United States." *Energy Research & Social Science* 55 (2019): 62-70.

Loi TS, Loo SL (2016) The impact of Singapore’s residential electricity conservation efforts and the way forward. Insights from the bounds testing approach. *Energy Policy* 98:735–743. <https://doi.org/10.1016/j.enpol.2016.02.045>

Yang Y, Deng YR, Chen FF (2018) Impact of population aging and industrial structure on CO2 Emissions and emissions trend prediction in China. *Atmos Pollut Res* 9:446–454. <https://doi.org/10.1016/j.apr.2017.11.008>

Zha, D., Liu, P. & Shi, H. Does population aging aggravate air pollution in China? *Mitigation and Adaptation Strategies for Global Change* 27, doi:10.1007/s11027-021-09993-y (2022).

Issue 11: Also the recommendation to activities indoors when there are high levels of pollutants can not work if the indoors ambients are contaminated with high levels of pollutants, as occurs with the use of solid fuels for cooking, for instance.

Response: Revised and clarified.

In the Discussion (*3.1 Population aging is the most important challenge*), we added relevant literature to remind our readers to strengthen the prevention of indoor PM_{2.5} air pollution (lines 278-282):

“However, indoor air pollution also needs to be considered, as the average adult in China spends around 86% of their time indoors³⁹. For example, Yun et al. pointed out that indoor pollution

caused by solid fuel use is far more serious than other activities, and promoting clean cooking and heating is an effective way to improve indoor air quality^{39,40}.”

In addition, we have explained our limitation of not considering indoor air pollution in the Discussion section. The specific contents are as follows (lines 334-340):

“Third, when estimating the health burden of air pollution, we considered both the effects of climate change and socioeconomic factors, but there are still some other factors that remain difficult to take into account. For example, we did not consider future policy interventions or indoor air pollution. Future public emergencies (e.g. the pandemic event) or possible policies (e.g. emissions reduction measures and temporary lockdowns) will affect the exposure of air pollution. Therefore, the projections in this study only reflect long-term trends in the health effects of outdoor PM_{2.5} air pollution.”

Part 4: Response to the reviewer 4

Issue 1: This paper makes forecasts of mortality attributable to PM_{2.5} exposure between 2017 and 2035 and decomposes the change in these deaths by the effects of changes in PM_{2.5} exposure, population growth, population ageing and mortality trends for causes affected by PM_{2.5}. I'll concentrate on commenting on the areas of my expertise: mortality and population estimation. The conclusion that ageing is the main driver of change in mortality attributable to PM_{2.5} is hardly surprising as the main diseases affected by PM_{2.5} have a steep age gradient and China's population is rapidly ageing.

Cause-specific mortality rates were taken from GBD 2017. A question is why not from the more recent version of GBD2019?

Response: Clarified.

We used the Cause-specific mortality rates from Yin et al¹⁴ at the provincial scale. As the Cause-specific mortality rates from GBD2019 did not consider the health disparities across provinces in China.

Issue 2: These mortality rates were forecast using a relatively crude method that was used in the earliest GBD forecasts. Looking at Figure 1 I have a number of questions/comments:

1. age-standardised (as stated in note, not in y-axis title; also no information on what standard was used) rates of IHD is forecast to be flat between 2017 and 2035 while stroke is forecast to continue a fast decline over the same period. I suspect this is an artefact of ignoring the pattern of a rise until 2010 and then a decline. As IHD and stroke share many risk factors, it is unlikely that this diverging pattern will actually occur. More recently in GBD forecasts are driven by forecasts of exposure to risk factors for the risk-dependent part of mortality rates and the older method used

in this paper is similar to what is being applied to the "risk-deleted" proportion of cause-specific mortality rates.

Response: Clarified and revised.

The age-standardized information has been added in the caption of the Figure 1.

In terms of the IHD forecast, after examining the data and methods, we still considered that the variation of stroke and IHD mortality in the original manuscript is reasonable. After searching related literature, we found that related studies also supported that the future trend of IHD mortality will differ from the trends of stroke in China, even though the two diseases share many risk factors. In terms of stroke, Ma et al. analyzed the trend of mortality rate of stroke in China from 1990 to 2019, and pointed out that the annual the age-standardized mortality rate in China showed a decreasing trend⁴¹:

“.....from 1990 to 2019, the age-standardised mortality rate decreased by 39·8% (28·6–50·7) and the DALY rate decreased by 41·6% (30·7–50·9)”;

and analyzed the possible causes⁴¹:

“.....we also found some improvement in the disease burden of stroke, which might be attributed to advancements in public stroke awareness and use of emergency medical services, improvement in medical treatment, as well as risk factor prevention of stroke.....We also found that the decrease in age-standardised DALYs attributable to stroke was mostly due to the decrease in YLLs (Years of Life Lost).”

In terms of IHD, Liu et al. studied the burden of IHD in China from 1990 to 2016, and pointed out that although the annual standardized mortality rate of stroke has begun to decrease in China, the mortality rate of IHD cannot be improved temporarily, and gave possible reasons⁴²:

“.....Factors contributing to these declines include improved health care coverage, upgraded medical technology, an improved public health environment on stroke prevention by the government..... However, despite these improvements, the age-standardized mortality rate for IHD has continued to increase. The China PEACE study reported that the admission rate for ST-segment elevation myocardial infarction grew rapidly from 2001 to 2011, but underuse of guideline-recommended therapies (eg, β -blockers and angiotensin-converting enzyme inhibitors) and use of therapies with unknown effectiveness remained common and had not significantly improved. Inadequate health care professional knowledge, structural inadequacies of care systems, and withdrawal from treatment at terminal status owing to affordability or cultural factors are also responsible for the increased IHD mortality rate.....”

Issue 3: I see no reference to forecasts of cause specific mortality being constrained by forecasts of total mortality rates. We tend to have greater faith in forecasts of all-cause mortality because data sources are less prone to measurement bias than those relying on a mix of physician-certified and coded deaths and verbal autopsy methods. This is an important step as separate forecasts cause by cause when aggregated by all causes can lead to large departures from the forecast of all deaths. I suspect the authors have not worried about this by forecasting just 6 causes of death.

Response: Revised and clarified.

We have emphasized in the *Methods* section that we only considered six causes to quantify the causes of DAPP as follows (lines 461-464):

“In addition, in the GBD2019, the quantification of DAPP required consideration of six diseases: chronic obstructive pulmonary disease, lower respiratory tract infection, lung cancer, ischemic heart disease, stroke, and type II diabetes mellitus. Specifically, we used the MR-BRT model developed and updated by GBD to estimate the deaths following Foreman et al⁶⁶.”

In addition, we also compared our estimates with other estimates using the total mortality rates. For example, some studies took all kinds of chronic diseases into account when quantifying DAPP. To verify the reliability of our results, we also compared our estimates with the predictions based on GEMM model (considering all chronic diseases and lower respiratory tract infections) and provided the comparison in the supplementary materials. We explained our comparison in the Discussion section. Although the absolute values of the two results are different, the trend of change is consistent. This also explains the rationality of the results of this study. The specific contents are as follows (lines 327-331):

“To assess the impact of this choice we compared the results of this study with deaths calculated by Geng et al⁵² based on the Global Exposure Mortality Model (GEMM), which models all-natural cause mortality and does not incorporate additional types of exposure, such as active smoking⁵³. Although the absolute values of the two results differ, the trends are consistent. Second, we assumed a constant association estimate for population vulnerability to PM_{2.5} air pollution.”

Issue 4: the panel on % elderly over 65 defies logic. How could the blue and yellow scenarios lead to such large differences? This would need to come from change in fertility, mortality or migration. In/out migration in China is small and unlikely to be a factor. Fertility could affect the denominator but would have to vary by an implausible amount to explain the differences in the panel. Similarly, mortality rates are unlikely to be able to create such a large variation in % 65+. I see similar change (in opposite direction by scenario) in total population. That makes me think there are large differences in assumptions on migration/fertility/mortality underlying these population estimates. However, the paper does not explain the methods of the population forecasts but refers to two publications. Ref 47 is a paper in a journal called *Climate Change Research* which I was unable to find online. Ref 48 has incomplete information (i.e. no journal listed) but I was able to find it. It seems that very different assumptions on fertility are the main reason for the variation. The high fertility assumptions in SSPs 2 and 3 have not (yet) been borne out after abandoning the one child policy, suggesting a return to much higher fertility is not so likely.

Response: Revised and clarified.

We are sorry that we used the wrong data to describe the proportion of older population in the mapping. Now we have corrected the data and re-mapped. The revised version is shown as follows (lines 128-129):

Figure 1. Projected change in driving factors from 2000-2035. (d-i) Age-standardized disease mortality. Note that the age structure of the sixth census in China was used as the standard age structure²⁸. We used 5-year smoothed time intervals to visualize the results.

The population data mainly came from Cai et al⁴⁴. We updated the references and explained the methods of population prediction in the Data section. The revised version is shown as follows (lines 408-416

“Cai et al³¹ considered the birth policy, household registration policy, and population ceiling of megacities implemented in China in recent years with a recursive multidimensional model. They also verified and compared the predicted total population and structured information results based on the latest statistical yearbook data of provinces and prefecture-level cities and the current gridded population data released by international institutions. Based on 2010 statistical data and the third Economic Census, Jiang et al⁶¹ estimated the labor force level, total factor productivity, and capital stock, and verified the parameters of the economic model. They also estimated China's economic progress under the SSPs using the improved Population, Development and Environment (PDE) model and Cobb-Douglas assumptions.”

Issue 5: the population forecasts are based on assumptions on all-cause mortality that are not consistent with those from which the cause specific mortality rates have been derived (GBD).

Response: Clarified.

Although the future total population is based on the assumption of all-cause mortality. Based on the comparable risk assessment framework, the DAPP are calculated separately for each age group and disease, and then summed to obtain the final DAPP. Therefore, the final results are still based on the assumption of disease-specific mortality.

Issue 6: Specific comments: Lines 25-27 you are not convincing me with this statement. Why?

Response: Revised and clarified.

We have rewritten the sentence (Estimating the health burden of air pollution in China against the background of population aging is of great significance for the well-being of elderly individuals and achieving the United Nations Sustainable Development Goals (SDGs)). The revised version is shown as follows (lines 26-29):

“Estimating the health burden of air pollution against the background of population aging is of great significance for achieving the United Nations Sustainable Development Goal 3.”

Issue 7: DAPP is an unusual acronym and best avoided

Response: Revised and clarified.

We tried to avoid using DAPP in the revised manuscript, but it strongly affected readability and word limit, especially in the Results and Methods sections. Some sentences would become ridiculously long. Therefore, we chose to keep using the acronym, DAPP, in the revised manuscript.

Issue 8: line 29: 'disease mortalities'; an awkward term; did you mean causes of death?

Response: Revised as suggested.

We've changed this to “causes of death”.

Issue 9: line 27-30: this sentence is a non-sequitur to me.

Response: Revised and clarified.

We have rewritten the sentence (Since previous studies did not fully consider the differences on drivers of deaths attributable to PM_{2.5} pollution (i.e., socioeconomic factors, climate factors, and disease mortalities) among provinces in China, the impact of future population aging on deaths attributable to PM_{2.5} pollution cannot be comprehensively and objectively estimated.), and the revised version is shown as follows (lines 29-33):

“Here, we estimated spatiotemporal changes in DAPP in China from 2000 to 2035 based on a risk assessment framework.....”

Issue 10: line 34: in an abstract at first mention of decomposition, I would expect to see the components of the decomposition

Response: Revised as suggested.

We've added the components of the decomposition in the *Abstract* (lines 33-37). The revised version is shown as follows (lines 32-33):

“.....and examined the drivers (i.e., population size, age structure of the population, PM_{2.5} concentration and related disease mortality) based on decomposition analysis.”

Issue 11: Line 38 (and repeater further on): with net decrease in PM_{2.5} attributable deaths in most scenarios, the word offset is misleading. What you mean is that ageing partially counters the effects of other factors (particularly lower forecasts of cause-specific mortality rates)

Response: Revised and clarified.

The previous expression was not accurate, so we have revised this sentence and other relevant places in the manuscript. The revised version is shown as follows (lines 36-37):

“We found that population aging will be the leading driver contributing to increased deaths attributable to PM_{2.5} air pollution and will counter the positive gains achieved by improvements in air pollution and healthcare.”

Issue 12: Line 39: that is a big statement to make about the need for improving health care. If you make this argument, I would like to see in much greater detail how you would envisage this being beneficial. Reducing exposure to PM_{2.5} would need to be the primary policy concern. Is your argument that we need to prevent the six diseases affected by PM_{2.5} by other means to make it less of a problem from PM_{2.5}? If so, what would you suggest?

Response: Revised and clarified.

We have rewritten this sentence. The revised version is shown as follows (line 38-39):

“Region-specific measures are required to mitigate the health burden of air pollution and this requires mutual cooperation and long-term efforts among regions in China.”

Issue 13: Line 47-48: ref 1 makes no mention of PM_{2.5} at all. Apart from that, if you make such a statement I would like to see more detail: where, by how much and over what period (not just saying 'recent').

Response: Revised and clarified.

We have rewritten the sentence (Recently, air pollution caused by particulate matter with a diameter of 2.5 µm or less (i.e., PM_{2.5}) has increased dramatically¹. Exposure to PM_{2.5} pollution has adverse effects on human health, such as premature death and pulmonary and cardiovascular disease), and the revised version is shown as follows (line 45-49):

“Fine particulate matter (PM_{2.5}) has become one of the most ubiquitous global air pollutants¹⁻³. Exposure to PM_{2.5} air pollution has adverse effects on human health, causing premature death via diseases such as pulmonary and cardiovascular disease^{4,5}. The number of deaths attributable to PM_{2.5} air pollution (DAPP) globally has also increased rapidly, from 1.14 million to nearly 3 million between 2000 and 2017⁶, becoming the fifth-leading mortality risk factor⁷.”

Issue 14: first paragraph of introduction: why discuss global issues in a paper that is about China?

Response: Revised and clarified

This is a description of the global context. We consider that the future trends of deaths attributable to PM_{2.5} air pollution in different regions of China will also have policy implications for other countries and regions in the world. We have explained it in *3.2 Insights on prevention*. Therefore, we consider it necessary to reveal the background of global aging in the introduction.

Issue 15: Line 68-69: medical conditions as an example of socio-economic development factors? mention of air pollution emissions is also odd as you are basically stating the obvious: 'DAPP' are affected by air pollution

Response: Revised and clarified.

To make our statement clearer, we have modified this section. In fact, based on the comparable risk assessment framework, the main core calculation method in this paper, both PM_{2.5} concentration and disease mortality are among the four major factors affecting deaths attributable to PM_{2.5} air pollution. The revised version is shown as follows (lines 63-70):

“The number of DAPP is determined by four factors: age structure, total population, PM_{2.5} concentration, and disease mortality⁷. Changes in these four factors are influenced by both climate change and socioeconomic change. For example, the changes in climate conditions such as temperature, humidity, and air pressure can directly affect the secondary transformation and diffusion of PM_{2.5}^{16,17}, thereby changing the atmospheric concentration of PM_{2.5}. Socio-economic development affects the emission of pollutants and the concentration of PM_{2.5} concentration, as well as the demographic characteristics and the level of available healthcare, which affect the proportion of older people in the population and the mortality rate of diseases^{18,19}.”

Issue 16: line 77: 'changes in medical conditions' Why mention this? Are you referring to trends in these conditions due to factors other than air pollution? If so, make that clearer.

Response: Revised and clarified.

To make the theme clearer, we have rewritten this paragraph focusing on previous studies related to population aging. The revised version is shown as follows (lines 70-88):

“Several studies have analyzed the relationship between PM_{2.5} air pollution and the older adult population in China, mainly based on historical data to analyze the impact of population aging on a specific disease²⁰⁻²² (e.g., respiratory disease or heart disease). For example, Chi et al. analyzed the impact of air pollution on the older patients with chronic obstructive pulmonary disease and found

that increased PM_{2.5} air pollution during the heating season was associated with a reduction in forced expiratory volume²². Several studies have analyzed historical relationships between population aging and DAPP. For example, Liu et al. estimated the number of deaths caused by air pollution and concluded that population aging caused an additional 950,000 deaths from 2004 to 2017, delaying the reduction of the health burden of PM_{2.5} air pollution²³. Although a few studies have estimated future changes of DAPP, with estimates based upon certain climatic or socioeconomic assumptions. For example, Yue et al. estimated the change of China's DAPP in 2030 under different PM_{2.5} concentration scenarios, but ignored changes in demographics and disease mortality²⁴. Wang et al. estimated the health burden of DAPP in 2020 and 2030 based on air quality scenarios and population trends under the Shared Socioeconomic Pathways (SSPs)²⁵. Although the study estimated the impact of population aging and other factors on DAPP in China based on future socioeconomic changes, it did not consider the changes in PM_{2.5} concentration in the context of climate change. Hence, there is an urgent need for a comprehensive and detailed quantification and projection of DAPP trends in China which ties these complex components together.”

Issue 17: line 86: Wu may not have used these but GBD produced province level estimates

Response: Revised and clarified.

Yes, GBD has already estimated deaths attributable to PM_{2.5} air pollution at the provincial level, but it estimates historical deaths attributable to PM_{2.5} air pollution in. Our paper estimated future dynamics in deaths attributable to PM_{2.5} air pollution in China. Moreover, the GBD study does not take into account the provincial differences in disease mortality in China. We obtained the provincial specific data from Dr. Peng Yin from China CDC and their studies¹⁴.

Related references:

Yin, P. et al. The effect of air pollution on deaths, disease burden, and life expectancy across China and its provinces, 1990-2017: an analysis for the Global Burden of Disease Study 2017. Lancet Planetary Health 4, E386-E398, doi:10.1016/s2542-5196(20)30161-3 (2020).

Issue 18: line 113 (and mentions elsewhere): the terms is comparative risk assessment, not comparable risk assessment

Response: Revised as suggested.

We have changed "comparable Risk Assessment" in the manuscript to "comparative Risk Assessment"

Issue 19: line 121 and following: you do not explain the various SSP scenarios. Many readers will not be familiar with that detail.

Response: Revised as suggested.

We have added the descriptions of SSP and RCP in the *Introduction* as follows (lines 92-99)

“Specifically, RCPs are multi-model global scenarios of greenhouse gas concentration trajectories. They use the radiative forcing per unit area at the end of the century to represent the climate impact of future greenhouse gas concentrations, covering a variety of climate mitigation levels and providing the basis for the simulation of PM_{2.5} in different climate change scenarios¹⁷. SSPs are socioeconomic development scenarios related to greenhouse gas emissions that take socioeconomic aspects such as economy, population, technology, lifestyle, policies, and institutions into account²⁶, making it possible to study the socioeconomic factors affecting DAPP.”

Meanwhile, in *Materials and methods*, we describe the changes of climate and socioeconomic factors under each scenario in more detail as follows (lines 371-389):

“ScenarioMIP classifies the scenarios of future socioeconomic and climate change into eight primary and secondary scenarios based on their priorities^{19,58}. In this study, we selected four scenarios with high priority in the first-level experiment to estimate DAPP. Among them, SSP1-2.6 is a sustainable green path characterized by low vulnerability and low mitigation challenges, with a mild climatic change and a small range of variation^{19,26}. In this scenario, the birth rate, death rate and migration rate are lower; population is more educated; and population is older¹⁶. SSP2-4.5 is an intermediate path with a moderate climatic change characterized by moderate social vulnerability and a moderate climatic change. In this scenario, social, economic and technological trends are similar to historical ones¹⁶, so that the changes in fertility, mortality, migration and the proportion of the older population also maintain current trends. SSP3-7.0 is a combined scenario of regional competing paths characterized by higher social vulnerability and higher mitigation challenges with a moderate to severe climatic change. In this scenario, population grows rapidly; the regional development is unbalanced, and the per capita economic and technological development level is low¹⁶. With rapid growth of population size, the fertility rate and mortality rate are relatively high, and the degree of population aging is relatively low²⁶. SSP5-8.5 is a scenario dominated by traditional fossil fuel combustion, characterized by high social vulnerability and high mitigation challenges, with a severe climatic change. In this scenario, social development will emphasize economic growth, while the fertility rate, mortality rate and migration rate are relatively low; and the degree of population aging is relatively high²⁶.”

Issue 20: Figure 2: presenting counts in units of ten thousands is rather unusual

Response: Revised as suggested.

We have changed the y axis to thousand as follows (lines 168-171):

Figure 2 Varying trends in deaths attributable to PM_{2.5} air pollution from 2000 to 2035. Note: changes in the total amount of DAPP (a) in China and among regions (c), and changes in DAPP per

capita (b) in China and among regions (d). The values for each region are the average estimates among the provinces in this region (see Table S2).

Issue 21: Line 251: aging the leading challenge? Over the forecast period of this paper, there is little to be done about that. Flagging it as a challenge implies you are able to do something about. I would argue that ability is rather limited.

Response: Revised as suggested.

Issue 22: Line 518: I presume you mean 1000 curve fits and not '1000th curve fit'

Response: Revised as suggested.

We have changed "1000th Curve fit" to "1000 curve fits".

Issue 23: line 551: first mention of GBDMAPS without any explanation or reference

Response: Revised as suggested.

We have added the explanation and references of GBDMAPS as follows (lines 342-343):

“To assess the impact of this choice we compared the results of this study with deaths calculated by Geng et al²⁵ based on the Global Exposure Mortality Model (GEMM)

Added reference:

Group GMW. Burden of disease attributable to coal-burning and other air pollution sources in China. Special report 2016; 20.

References:

- Bu, X. et al. Global PM2.5-attributable health burden from 1990 to 2017: Estimates from the Global Burden of disease study 2017. *Environmental Research* 197, doi:10.1016/j.envres.2021.111123 (2021).
- Group, G. M. W. Burden of Disease Attributable to Coal-Burning and Other Air Pollution Sources in China, <<https://www.healtheffects.org/publication/burden-disease-attributable-coal-burning-and-other-air-pollution-sources-china>> (2016).
- Collaborators, G. R. F. Global burden of 87 risk factors in 204 countries and territories, 1990-2019: a systematic analysis for the Global Burden of Disease Study 2019. *The Lancet* 396, 1223-1249, doi:10.1016/s0140-6736(20)30752-2 (2020).
- Xie, Y., Dai, H., Dong, H., Hanaoka, T. & Masui, T. Economic Impacts from PM2.5 Pollution-Related Health Effects in China: A Provincial-Level Analysis. *Environmental Science & Technology* 50, 4836-4843, doi:10.1021/acs.est.5b05576 (2016).
- Southerland, V. A. et al. Global urban temporal trends in fine particulate matter (PM2.5) and attributable health burdens: estimates from global datasets. *The Lancet. Planetary health* 6, e139-e146, doi:10.1016/s2542-5196(21)00350-8 (2022).
- Collaborators, G. S. Measuring progress from 1990 to 2017 and projecting attainment to 2030 of the health-related Sustainable Development Goals for 195 countries and territories: a systematic analysis for the Global Burden of Disease Study 2017. *The Lancet* 392, 2091-2138,

- doi:10.1016/s0140-6736(18)32281-5 (2018).
- Cohen, A. J., Brauer, M. & Burnett, R. Estimates and 25-year trends of the global burden of disease attributable to ambient air pollution: an analysis of data from the Global Burden of Diseases Study 2015 (vol 389, pg 1907, 2017). *Lancet* 391, 1576-1576, doi:10.1016/s0140-6736(18)30900-0 (2018).
 - Forouzanfar, M. H. et al. Global, regional, and national comparative risk assessment of 79 behavioural, environmental and occupational, and metabolic risks or clusters of risks in 188 countries, 1990-2013: a systematic analysis for the Global Burden of Disease Study 2013. *Lancet* 386, 2287-2323, doi:10.1016/s0140-6736(15)00128-2 (2015).
 - Murray, C. J. L., Ezzati, M., Lopez, A. D., Rodgers, A. & Vander Hoorn, S. Comparative quantification of health risks conceptual framework and methodological issues. *Population health metrics* 1, 1, doi:10.1186/1478-7954-1-1 (2003).
 - Yue, H., He, C., Huang, Q., Yin, D. & Bryan, B. A. Stronger policy required to substantially reduce deaths from PM_{2.5} pollution in China. *Nature Communications* 11, doi:10.1038/s41467-020-15319-4 (2020).
 - Wang, Q., Wang, J., Zhou, J., Ban, J. & Li, T. Estimation of PM_{2.5}-associated disease burden in China in 2020 and 2030 using population and air quality scenarios: a modelling study. *Lancet Planetary Health* 3, E71-E80, doi:10.1016/s2542-5196(18)30277-8 (2019).
 - Jiao, K., Xu, M. & Liu, M. Health status and air pollution related socioeconomic concerns in urban China. *International Journal for Equity in Health* 17, doi:10.1186/s12939-018-0719-y (2018).
 - Maji, K. J., Ye, W.-F., Arora, M. & Nagendra, S. M. S. PM_{2.5}-related health and economic loss assessment for 338 Chinese cities. *Environment International* 121, 392-403, doi:10.1016/j.envint.2018.09.024 (2018).
 - Yin, P. et al. The effect of air pollution on deaths, disease burden, and life expectancy across China and its provinces, 1990-2017: an analysis for the Global Burden of Disease Study 2017. *Lancet Planetary Health* 4, E386-E398, doi:10.1016/s2542-5196(20)30161-3 (2020).
 - Beard, J. R. et al. The World report on ageing and health: a policy framework for healthy ageing. *Lancet* 387, 2145-2154, doi:10.1016/s0140-6736(15)00516-4 (2016).
 - Fang, D. et al. Clean air for some: Unintended spillover effects of regional air pollution policies. *Science Advances* 5, doi:10.1126/sciadv.aav4707 (2019).
 - Gidden, M. J., Riahi, K., Smith, S. J., Fujimori, S. & Takahashi, K. J. G. M. D. D. Global emissions pathways under different socioeconomic scenarios for use in CMIP6: a dataset of harmonized emissions trajectories through the end of the century. 1-42 (2018).
 - Rafaj, P., Kiesewetter, G., Krey, V., Schpp, W. & Vuuren, D. J. E. R. L. Air quality and health implications of 1.5–2°C climate pathways under considerations of ageing population: A multi-model scenario analysis. (2021).
 - O'Neill, B. C. et al. The roads ahead: Narratives for shared socioeconomic pathways describing world futures in the 21st century. *Global Environmental Change-Human and Policy Dimensions* 42, 169-180, doi:10.1016/j.gloenvcha.2015.01.004 (2017).
 - Cai, W., Li, K., Liao, H., Wang, H. & Wu, L. Weather conditions conducive to Beijing severe haze more frequent under climate change. *Nature Climate Change* 7, 257-+, doi:10.1038/nclimate3249 (2017).
 - Turnock, S. T., Allen, R. J., Andrews, M., Bauer, S. E. & Zhang, J. Historical and future

- changes in air pollutants from CMIP6 models. (2020).
- Jerrett, M. et al. Comparing the Health Effects of Ambient Particulate Matter Estimated Using Ground-Based versus Remote Sensing Exposure Estimates. *Environmental Health Perspectives* 125, 552-559, doi:10.1289/ehp575 (2017).
 - Chowdhury, S., Dey, S. & Smith, K. R. Ambient PM_{2.5} exposure and expected premature mortality to 2100 in India under climate change scenarios. *Nature Communications* 9, doi:10.1038/s41467-017-02755-y (2018).
 - China, N. B. o. S. i. The sixth census in China, <<http://www.stats.gov.cn/tjsj/pcsj/rkpc/6rp/indexch.htm>> (2010).
 - Geng, G. N. et al. Drivers of PM_{2.5} air pollution deaths in China 2002-2017. *Nature Geoscience* 14, 645+, doi:10.1038/s41561-021-00792-3 (2021).
 - Burnett, R. & Cohen, A. Relative Risk Functions for Estimating Excess Mortality Attributable to Outdoor PM(2.5)Air Pollution: Evolution and State-of-the-Art. *Atmosphere* 11, doi:10.3390/atmos11060589 (2020).
 - Murray, C. J. L. & Lopez, A. D. On the comparable quantification of health risks: Lessons from the Global Burden of Disease Study. *Epidemiology* 10, 594-605, doi:10.1097/00001648-199909000-00029 (1999).
 - Rizmie, D., de Preux, L., Miraldo, M. & Atun, R. Impact of extreme temperatures on emergency hospital admissions by age and socio-economic deprivation in England. *Social Science & Medicine* 308, doi:10.1016/j.socscimed.2022.115193 (2022).
 - Yang, Z. M., Wang, Q. & Liu, P. F. Extreme temperature and mortality: evidence from China. *International Journal of Biometeorology* 63, 29-50, doi:10.1007/s00484-018-1635-y (2019).
 - Yin, H. et al. Population ageing and deaths attributable to ambient PM_{2.5} a of economic cost. *Lancet Planetary Health* 5, E356-E367 (2021).
 - Zha, D., Liu, P. & Shi, H. Does population aging aggravate air pollution in China? Mitigation and Adaptation Strategies for Global Change 27, doi:10.1007/s11027-021-09993-y (2022).
 - Chi, R. et al. Different health effects of indoor- and outdoor-originated PM_{2.5} on cardiopulmonary function in COPD patients and healthy elderly adults. *Indoor Air* 29, 192-201, doi:10.1111/ina.12521 (2019).
 - Liu, Y. et al. Population aging might have delayed the alleviation of China's PM_{2.5} health burden. *Atmospheric Environment* 270, doi:10.1016/j.atmosenv.2021.118895 (2022).
 - Silva et al. Future global mortality from changes in air pollution attributable to climate change (vol 7, pg 647, 2017).
 - Guan, D. et al. The socioeconomic drivers of China's primary PM_{2.5} emissions. *Environmental Research Letters* 9, doi:10.1088/1748-9326/9/2/024010 (2014).
 - Jones, B., Tebaldi, C., O'Neill, B. C., Oleson, K. & Gao, J. Avoiding population exposure to heat-related extremes: demographic change vs climate change. *Climatic Change* 146, 423-437, doi:10.1007/s10584-017-2133-7 (2018).
 - Mistry, M. N. A High Spatiotemporal Resolution Global Gridded Dataset of Historical Human Discomfort Indices. *Atmosphere* 11, doi:10.3390/atmos11080835 (2020).
 - Yuan xiaochen, Y. b., Wei siyi, Yang zhiming. Assessment of the impact of global warming on labor health in China. (2022).
 - Li, L., Zheng, Y. & Ma, S. Indoor Air Purification and Residents' Self-Rated Health: Evidence from the China Health and Nutrition Survey. *International Journal of Environmental Research*

and Public Health 19, doi:10.3390/ijerph19106316 (2022).

- Yun, X. et al. Residential solid fuel emissions contribute significantly to air pollution and associated health impacts in China. *Science Advances* 6, doi:10.1126/sciadv.aba7621 (2020).
- Ma, Q. et al. Temporal trend and attributable risk factors of stroke burden in China, 1990-2019: an analysis for the Global Burden of Disease Study 2019. *Lancet Public Health* 6, E897-E906 (2021).
- Liu, S. W. et al. Burden of Cardiovascular Diseases in China, 1990-2016 Findings From the 2016 Global Burden of Disease Study. *Jama Cardiology* 4, 342-352, doi:10.1001/jamacardio.2019.0295 (2019).
- Foreman, K. J. et al. Forecasting life expectancy, years of life lost, and all-cause and cause-specific mortality for 250 causes of death: reference and alternative scenarios for 2016-40 for 195 countries and territories. *Lancet* 392, 2052-2090, doi:10.1016/s0140-6736(18)31694-5 (2018).
- Chen, Y. et al. Provincial and gridded population projection for China under shared socioeconomic pathways from 2010 to 2100. *Scientific Data* 7, doi:10.1038/s41597-020-0421-y (2020).
- Jiang, T. et al. Estimated population changes in China and provinces under the IPCC shared socio-economic pathway. *Climate Change Research* 013, 128-137 (2017).

Reviewer comments, second round

Reviewer #1 (Remarks to the Author):

Thank you for considering my comments. I am satisfied with the authors' responses and their amendments to the manuscript.

Reviewer #2 (Remarks to the Author):

I would like to thank the authors for their thorough revision of the manuscript. I believe that the quality of the work has improved and most of my suggestions/comments were properly addressed. I just would like to point out a few minor points:

- Abstract: when reporting the main results, it is difficult for the reader understand the magnitude of the increased mortality if no reference is provided - that is, what means for China an increase in 1.9 thousand deaths?
- In Section 2.3, I would suggest briefly introducing the plot, so the reader can follow the explanation.
- I would replace the term "elderly" with older adults (or population), the latter is preferred in gerontology

Reviewer #3 (Remarks to the Author):

Dear authors,

I recognized all the work in answering the questions and suggestions by the reviewers.

I consider the authors have answered them resulting in an improved manuscript and recommend its acceptance.

Reviewer #4 (Remarks to the Author):

Review of RTR (numbers refer to those in RTR) by reviewer 4:

1 GBD2019 results have been made for China provinces. As you state getting GBD2017 estimates by province through China CDC, can you not request the more recent estimates? Note that these results are not publicly available as stipulated by an agreement with China CDC

2 After 2013 IHD mortality rates in China have been declining. This declining trend is confirmed in the latest round of GBD estimates which will become available in early 2023. By drawing a simple linear regression from 1990 onwards, your forecasts ignore the recent trend and therefore you are likely overstating mortality rates from IHD in your forecasts

3 Forecasting just 6 cause of mortality does not take away the problem we often see that forecasts of individual diseases can vary significantly from forecasts of total mortality which we tend to trust more as we tend to have more and stronger evidence in making those estimates. Even if you are only interested in 6 causes, you could have created a rest category of all other causes and made separate forecasts for it and then constrain all 7 forecasts to the all-cause mortality 'envelope'

5 Your response misses the point I was making. All-cause mortality rates are an input into population forecasts. GBD cause-specific mortality estimates are constrained to a total of all-cause mortality estimated in GBD. I was remarking that there is an inconsistency between the population estimates you use and what is used in GBD which influences the estimates you are using. I don't think it is a major issue but possibly something to flag in limitations.

6 In the rewrite of this sentence, it still seems a stretch to state that your estimates will be of great significance for achieving SDG3 unless you expand on through what mechanism these estimates will trigger some action that will contribute to achieving SDG3

14 At best, that can be picked up in discussion. It seems out of place in this paper

Response to the Editor and Reviewers

We would like to express our gratitude to the editor and anonymous reviewers for their valuable comments and suggestions for improving the quality of the paper. We have carefully considered all the points raised by them. We are providing detailed point-by-point responses to all questions and recommendations by the reviewers. In the responses below, red fonts are the revised texts.

Part 1: Response to the reviewer 1

Thank you for considering my comments. I am satisfied with the authors' responses and their amendments to the manuscript.

Response: Thank you again for your help in improving the quality of the manuscript.

Part 2: Response to the reviewer 2

I would like to thank the authors for their thorough revision of the manuscript. I believe that the quality of the work has improved and most of my suggestions/comments were properly addressed. I just would like to point out a few minor points:

Issue 1: Abstract: when reporting the main results, it is difficult for the reader understand the magnitude of the increased mortality if no reference is provided - that is, what means for China an increase in 1.9 thousand deaths?

Response: Revised.

We have revised the absolute values to relative values when reporting the main results. We also compare the relative value the SDG 3.9 target. The revised version is shown as follows (lines 35-39):

“.....deaths were projected to decrease 24.6% (-3.8%-40.0%, 95% CI), 2.6% (-14.9%-9.5%) and 18.3% (24.7%-31.0%) under the SSP1-2.6, SSP2-4.5 and SSP5-8.5 scenario, respectively, but increase 8.4% (1.3%-14.9%) under SSP3-7.0 scenario, which suggests that it is difficult to achieve Sustainable Development Goal 3.9 in China.”

Issue 2: In Section 2.3, I would suggest briefly introducing the plot, so the reader can follow the explanation.

Response: Revised as suggested.

We have briefly introduced the plot in section 2.3. The revised version is shown as follows (lines 180-182):

“Figure 3 shows the contribution of the four factors (i.e., age structure, total population, air quality, and disease mortality) on the changes in DAPP between 2017 and 2025, and between 2025 and 2035, respectively.”

Issue 3: I would replace the term "elderly" with older adults (or population), the latter is preferred in gerontology

Response: Revised as suggested.

We have replaced the term "elderly" with older adults (or older population) throughout the manuscript.

Part 3: Response to the reviewer 3

Dear authors,

I recognized all the work in answering the questions and suggestions by the reviewers.

I consider the authors have answered them resulting in an improved manuscript and recommend its acceptance.

Response: Thank you again for your help in improving the quality of the manuscript.

Part 4: Response to the reviewer 4

Review of RTR (numbers refer to those in RTR) by reviewer 4:

Issue 1: GBD2019 results have been made for China provinces. As you state getting GBD2017 estimates by province through China CDC, can you not request the more recent estimates? Note that these results are not publicly available as stipulated by an agreement with China CDC

Response: Clarified

We do not have the access to the most recent estimates by province through China CDC because our collaboration with China CDC only allowed us to have the access to the GBD 2017 estimates.

Issue 2: After 2013 IHD mortality rates in China have been declining. This declining trend is confirmed in the latest round of GBD estimates which will become available in early 2023. By drawing a simple linear regression from 1990 onwards, your forecasts ignore the recent trend and therefore you are likely overstating mortality rates from IHD in your forecasts

Response: Revised and clarified.

We have added more details about the trend of the mortality rates in the *Discussion* section. The revised version is shown as follows (lines 347-348):

“Fourth, the forecasts of mortality rates may ignore the declining trend of IHD mortality rate after 2013 and therefore the DAPP cause by IHD may be overestimated⁵⁵.”

Issue 3: Forecasting just 6 causes of mortality does not take away the problem we often see that forecasts of individual diseases can vary significantly from forecasts of total mortality which we tend to trust more as we tend to have more and stronger evidence in making those estimates. Even if you are only interested in 6 causes, you could have created a rest category of all other causes and made separate forecasts for it and then constrain all 7 forecasts to the all-cause mortality ‘envelope’

Response: Revised as suggested.

We have created a category of the all other causes and made separate forecasts for it and then constrain all forecasts to the all-cause mortality ‘envelope’. The details are shown as follows (lines 483-510):

“ Since estimating just 6 causes of mortality may vary from the estimates of total mortality from all causes, we constrained our estimates of DAPP from the 6 causes to the all-cause mortality envelope at the national level published by GBD 2017 (Figure S3). First, according to the historical mortality rates of all-cause (level 1 causes) at the national level provided by GBD2017, we estimated the mortality rates of all-cause diseases during 2017-2035 based on equation (7). These estimates formed the “envelope” to constrain the level 2 causes (Figure S3). Second, we estimated the mortality rates of diseases at level 2, i.e., non-communicable diseases (NCD), communicable, maternal, neonatal and nutritional disease (CMNND) and injuries during 2017-2035 based on equation (7), according to historical data provided by GBD2017. We then constrained the mortality rates of the three level 2 diseases to the “envelope” of all causes (level 1 causes) as follows,

$$Rate_{1stconstrained,a,y} = \frac{Rate_{NCD,y} + Rate_{CMNND,y} + Rate_{injuries,y}}{Rate_{all,y}} \times Rate_{a,y} \quad (9)$$

where $Rate_{1stconstrained,a,y}$ is the constrained mortality rate for the disease a in year y . $Rate_{NCD,y}$, $Rate_{CMNND,y}$, $Rate_{injuries,y}$, and $Rate_{all,y}$ are the original mortalities rates estimated for NCD, CMNND, injuries and all causes based on equation (7), respectively. $Rate_{a,y}$ is the original mortality rate of disease a in year y calculated by equation (7). Third, we further constrained the mortality rates of the six diseases at level 3 to the envelope of the constrained level 2 mortality rates above in a similar way. Taking NCDs as an example, we constrained the mortalities rates of the five level 3 causes (COPD, DM2, IHD, LC, and stroke) and the mortality rates of the other causes (rest-NCD for short) to the constrained level 2 mortality rates (i.e., $Rate_{1stconstrained,a,y}$) as below,

$$Rate_{2ndconstrained,n} = \frac{\sum Rate_{n,y} + Rate_{rest-NCD,y}}{Rate_{1stconstrained,NCD,y}} \times Rate_{n,y} \quad (10)$$

where $Rate_{2ndconstrained,n}$ is the constrained mortality rate of a level 3 disease n in year y . $Rate_{n,y}$ represents the original mortality rate of a level 3 disease n in year y estimated by equation (7). $Rate_{n,y}$ and $Rate_{rest-NCD,y}$ are the mortalities rates of a level 3 NCD estimated by equation (7) using historical data from Yin et al²⁵, and the mortality rate of rest NCDs estimated by equation (7) based upon GBD2017, respectively. The constrained mortality rate of LRI was calculated in a similar way. After two rounds of the constraining process, our estimates for the 6 causes of mortality are within the range of all-cause mortality, and are consistent with GBD2017.

In addition to the six diseases above (five non-communicable diseases and one communicable, maternal, neonatal and nutritional disease) relative to DAPP, we created a rest category of all other causes, including the rest Non-communicable diseases (rest-NCD, for short) and the rest communicable, maternal, neonatal and nutritional disease (rest-CMNND, for short). In addition, according to the classification of GBD2017, as all causes can be divided into three categories: infectious diseases, noncommunicable diseases, and injuries (Fig.S3), before constraining the six disease mortality relative to DAPP, we first constrained the mortality at level2, and then constrained the disease mortality at level 3 based on the constrained results at level 2. ”

Figure S3 the diseases considering in this paper

In the revised manuscript, we focused on reporting the 6 causes of mortality as they are more robust at the provincial level. We also updated all the figures, tables and numbers in the main texts and supplementary files after the new calculations.

Issue 4: Your response misses the point I was making. All-cause mortality rates are an input into population forecasts. GBD cause-specific mortality estimates are constrained to a total of all-cause mortality estimated in GBD. I was remarking that there is an inconsistency between the population estimates you use and what is used in GBD which influences the estimates you are using. I don't think it is a major issue but possibly something to flag in limitations.

Response: Revised as suggested.

We have added this issue in the section of *limitations*, considering the population forecasts are based on assumptions on all-cause mortality that are not consistent with those from which the cause specific mortality rates have been derived in GBD. The revised version is shown as follows lines(348-351):

“Also, the population forecasts we used in this study are based on assumptions on all-cause mortality that are not consistent with those from which the cause specific mortality rates have been derived in the GBD, which might also influence our estimates in DAPP.”

Issue 5 (a follow-up question for former issue 6): In the rewrite of this sentence (“*Estimating the health burden of air pollution against the background of population aging is of great significance for achieving the United Nations Sustainable Development Goal 3.*”), it still seems a

stretch to state that your estimates will be of great significance for achieving SDG3 unless you expand on through what mechanism these estimates will trigger some action that will contribute to achieving SDG3.

statement. Why?

Response: We have added some more details about the Sustainable Development Goals. The revised version is shown as follows (lines 61-66):

“..... establishing a pre-warning system for reducing exposure to heavily polluted air, increasing financial investment to alleviate their economic burden for treating related diseases. The insights gained can help other countries deal with the health burden of air pollution in the context of population aging and promote progress towards SDG 3.9.1 (reducing mortality rate attributed to household and ambient air pollution) and SDG 11.6.2 (reducing annual mean levels of fine particulate matter).”

Issue 6: At best, that can be picked up in discussion. It seems out of place in this paper.

A follow-up question for former issue 14 (Issue 14: first paragraph of introduction: why discuss global issues in a paper that is about China?)

Response: We have removed these texts at the global scale in the *Introduction* section.

Reviewer comments, third round

Reviewer #4 (Remarks to the Author):

You still have not adequately addressed my first two concerns. With regard to access to GBD 2019 data: are you saying that China CDC is preventing you from accessing the data?or did you not try?

The more important one is about the IHD trends. It makes up a large part of your forecasts and, thus, continuing to assume an upward trend in IHD mortality while in fact these have been declining since 2013 means that you are overstating your forecasts. Adding a line in limitations around this issue is not enough

Response to the Editor and Reviewers

We would like to express our gratitude to the editor and anonymous reviewers for their valuable comments and suggestions for improving the quality of the paper. We have carefully considered all the points raised by them. We are providing detailed point-by-point responses to all questions and recommendations by the reviewers. In the responses below, red fonts are the revised texts.

Part 1: Response to the reviewer 4

Issue 1: You still have not adequately addressed my first two concerns. With regard to access to GBD 2019 data: are you saying that China CDC is preventing you from accessing the data?or did you not try?

Response: Revised as suggested.

We have contacted China CDC and got the GBD2019 at the provincial level. Then, we re-conducted all the calculation and revised the texts, tables and figures accordingly. The trends of estimated deaths under the four scenarios remained and the main conclusions remained as “population aging will be the leading contributor to increased deaths attributable to PM2.5 air pollution”. Also, we used the new GBD 2019 at the national level to constrain our estimates of DAPP from the 6 causes to the all-cause mortality envelope, instead of GBD2017. The revised texts in the manuscript are marked as blue font.

Issue 2: The more important one is about the IHD trends. It makes up a large part of your forecasts and, thus, continuing to assume an upward trend in IHD mortality while in fact these have been declining since 2013 means that you are overstating your forecasts. Adding a line in limitations around this issue is not enough.

Response: Resolved.

After using the new dataset from China CDC, the IHD trend becomes declining in the future. Please check the revised Figure 1f for the projected change in mortality of IHD.

Figure 1. Projected change in driving factors from 2000-2035. (d-i) Age-standardized disease mortality. Note that the age structure of the sixth census in China was used as the standard age structure²⁷. We used 5-year smoothed time intervals to visualize the results.

Reviewer comments, fourth round

Reviewer #4 (Remarks to the Author):

Thank you for updating the manuscript to use most recently available data by CHina provinces and to take into account the recent declines in IHD mortality rather than the previous rather large increases.